# FROM INTERVENTION TO DOMAIN TRANSPORTATION: A NOVEL PERSPECTIVE TO OPTIMIZE RECOMMENDATION

**Da Xu**
Walmart Labs
Sunnyvale, CA 94086, USA
DaXu5180@gmail.com

**Yuting Ye**
Division of Biostatistics
University of California, Berkeley
Berkeley, CA 94720, USA
yeyt@berkeley.edu

**Chuanwei Ruan**
Instacart
San Francisco, CA 94107, USA
Ruanchuanwei@gmail.com

## ABSTRACT

The interventional nature of recommendation has attracted increasing attention in recent years. It particularly motivates researchers to formulate learning and evaluating recommendation as causal inference and data missing-not-at-random problems. However, few take seriously the consequence of violating the critical assumption of overlapping, which we prove can significantly threaten the validity and interpretation of the outcome. We find a critical piece missing in the current understanding of information retrieval (IR) systems: as interventions, recommendation not only affects the already observed data, but it also interferes with the target domain (distribution) of interest. We then rephrase optimizing recommendation as finding an intervention that best transports the patterns it learns from the observed domain to its intervention domain. Towards this end, we use domain transportation to characterize the learning-intervention mechanism of recommendation. We design a principled transportation-constraint risk minimization objective and convert it to a two-player minimax game. We prove the consistency, generalization, and excessive risk bounds for the proposed objective, and elaborate how they compare to the current results. Finally, we carry out extensive real-data and semi-synthetic experiments to demonstrate the advantage of our approach, and launch online testing with a real-world IR system.

## 1 INTRODUCTION

For information retrieval (IR) systems, the users' willingness to interact is often intervened by the content we show. Recommendations not only impact the potential response of the users, but also change the nature of the data collected for training machine learning models (Bottou et al., 2013). The interventional nature of recommendation, as well as the fact that many questions in IR are counterfactual, e.g., "what the response would have been if we recommended something else", makes it natural to rephrase recommendation in the context of causal inference or missing data problems (Rubin, 2005; Little et al., 2002). Among the two disciplines, technical tools of importance weighting (IW) and domain adaptation (DA) have been commonly applied. Recently, there has been widespread interest in adapting those tools for learning the best recommendation (Schn-

abel et al., 2016; Bonner and Vasile, 2018), but we find a critical discussion missing from most existing literature in this direction. In many scientific disciplines, the effectiveness of IW and DA relies heavily on overlapping between source and target domain (Austin and Stuart, 2015; David et al., 2010). The underlying reason also happens to be the motivation of exploration-exploitation techniques including bandits and reinforcement learning (Auer et al., 2002; Sutton and Barto, 1998): in the less-explored regions where few observations are collected, the confidence of selecting any hypothesis is discounted.

Unfortunately, modern IR systems are often cautious about exposing users to less relevant content for the sake of immediate satisfaction and revenue, even with the help of bandits or RL (Bonner and Vasile, 2018). In particular, many real-world recommenders are deterministic, e.g., an item is shown with probability either zero or one, which further limits the exploration and thus the coverage of collected data. In Section 3, we rigorously prove for both IW- and DA-based learning the hardness or impossibility results caused by insufficient overlap. Intuitively, when a less-explored instance is passed to a candidate hypothesis, even if DA or IW can improve the accuracy of the prediction, they inevitably increase the uncertainty of that prediction as a consequence of the bias-variance tradeoff. To a certain point, the increased uncertainty from having too many instances in the weakly overlapped regions will fundamentally limit what any learning procedure can achieve. Further, if the instance lies in the non-overlapped region, we can barely exceed regular extrapolation[1] (Xu et al., 2021b).

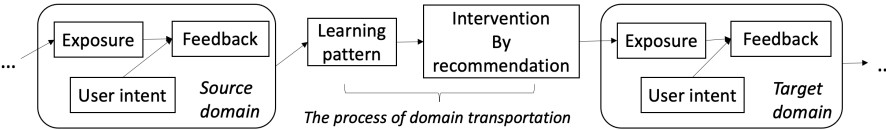

Figure 1: A realistic working mechanism of IR systems that emphasize domain transportation.

We resolve the fundamental limitation of insufficient overlapping in IR with the key insight that the interventional impact of recommendation is not exclusive to observed data, but also interferes with the target domain of interest — the instances that will be affected by the recommendations we are about to make. For instance, the placement of a recommendation on the webpage proposed to the user directly influences the occurrence of clicks. Therefore, we alternatively consider optimizing recommendation as searching for an intervention that best transports the patterns it learns from the source domain to its intervention domain. The procedure is best described by transport instead of transfer because it will interfere with how the learnt patterns are carried to the target domain. This change of view describes a complex but realistic working mechanism that involves learning, intervention, and transportation, as depicted in Figure 1. On the one hand, we wish to learn patterns that can be transported to the target domain of interest. On the other hand, the patterns we identify lead to the intervention (in the form of recommendation) that generates the target domain. Following this working mechanism, we impose constraints on the domain transportation between the (reweighted) source domain and the future intervention domain. We point out that reweighting is essential for learning from interventional feedback data (Rosenbaum and Rubin, 1983), especially for the working mechanism we identify here because the relationship between the source and target domain is more involved. Towards this end, we propose a novel solution via transportation-constraint risk minimization. We rigorously investigate for our objective:

- the consistency, generalization, and excessive risk bounds that provide theoretical insights and guarantees;

- the equivalence to a two-player minimax game that can be efficiently solved by gradient descent-ascent algorithm;

- convergence to the standard IW-based solution for the ideal fully-overlapped bandit feedback;

We demonstrate the empirical performance of our method via comprehensive experiments via simulation, real-data analysis, as well as online experimentation and testing in a real-world IR system.

---

[1]The presumption for this argument is that the underlying causal mechanism is unknown, which applies to the settings we discuss in the paper.

## 2 RELATED WORK

Bottou et al. (2013) first reveals the importance of characterizing the working mechanism of IR systems. However, they study specifically the advertising systems where the underlying causal structures are assumed given. Our work significantly extends their discussion by moving beyond causal mechanisms. Issues caused by ignoring the interventional nature of recommendation have been pointed out for numerous scenarios (Chen et al., 2020). Steck (2010) and Liang et al. (2016) address optimizing recommendation in the context of missing data and causal inference, and the majority of recent follow-up work investigate particularly inverse propensity weighting (IPW) and learning domain invariant representation methods (Chen et al., 2020; Saito, 2020; Yang et al., 2018; Joachims et al., 2017; Bonner and Vasile, 2018; Agarwal et al., 2019; Xu et al., 2020). As mentioned in Section 1, they largely ignore the overlapping issue even when the recommendation is deterministic. Our theoretical results on the interplay between overlapping and learning performance are novel and add to the current venue of revealing the tradeoff for IW- and DA-based learning (Byrd and Lipton, 2019; Johansson et al., 2019; Ben-David and Urner, 2012). See Appendix D for further discussions on related work.

Existing solutions for insufficient overlapping focus primarily on reducing or calibrating the variance, e.g., as discussed in Swaminathan and Joachims (2015b); Wang et al. (2020); Johansson et al. (2020). They do not account for the unique working mechanism of IR where the domain discrepancy can be actively controlled. We devise the domain-transportation constraint using the Wasserstein distance originated from the optimal transport theory (Courty et al., 2016; Redko et al., 2017). The idea of converting a Wasserstein-constraint objective to two-player minimax game is similar to that of the generative adversarial network (Gulrajani et al., 2017), however, we do not make any generative assumption on the observed data. Finally, *learning from bandit feedback* and the associated *counterfactual risk minimization* can be viewed as a special case of our framework (Swaminathan and Joachims, 2015a). We will use it as a test bed to examine the guarantees of the proposed approach.

## 3 PRELIMINARIES

We use upper-case letters to denote random variables and measures, bold-font letters for vectors and matrices. We denote by $\|\cdot\|_2, \|\cdot\|_\infty, \|\cdot\|_L$ the $\ell_2, \ell_\infty$ norm and the Lipschitz constant of a function.

For clarity, we borrow the item recommendation setting to illustrate the IR system we discuss next. The set of all users and items are given by $\mathcal{U}$ and $\mathcal{I}$. To be concise, we denote the features (or pre-trained embeddings) for users and items by $\mathbf{x}_u$ for $u \in \mathcal{U}$ and $\mathbf{x}_i$ for $i \in \mathcal{I}$. We study primarily the practical implicit-feedback setting with $Y_{ui} \in \{0, 1\}$, e.g., users express preferences implicitly via clicks. Let $\mathcal{D}$ be the population of all possible $(\mathbf{x}_u, \mathbf{x}_i)$ or $(u, i)$ depending on the context. The embeddings can also be free parameters under the representation mapping $f$ from $\mathcal{X}$ or $\mathcal{U} \cup \mathcal{I}$ to $\mathbb{R}^d$. We let $P$ be *any* base probability measure supported on the subsets of $\mathcal{D}$, and let $P_n$ be the empirical versions according to the collected feedback of $\mathcal{D}_n$ where $n$ is the sample size. We assume the weight of a user-item is positive and bounded by a constant $M$. Equipped with either individual weights $w_{ui} \in [0, M]$ or weighting function $w : \mathcal{U} \times \mathcal{I}$ (or $\mathcal{X} \times \mathcal{X}$) $\rightarrow [0, M]$, the reweighted measures are given by such as: $P_w(u, i) \propto w(u, i) \cdot P(u, i)$.

The candidate recommendation algorithm $f \in \mathcal{F}$ outputs a score or probability for each user-item pair. Top-$k$ recommender often ranks the output and then renders the top-$k$ items to the user. We denote this procedure by the function: $\text{reco}(u, i; f) \in \{0, 1\}$. Define the new measure on $\mathcal{D}$ induced by deploying $f$ on $\mathcal{D}$ as: $P_f(u, i) \propto \text{reco}(u, i; f) \cdot P(u, i)$. If we choose the base measure $P$ as uniform, it is clear that $P_f$ is a uniform measure on the $(u, i)$ pairs that *will be exposed*: $\{(u, i) \in \mathcal{D} : \text{reco}(u, i; f) = 1\}$. Finally, given a loss function $\ell$ for $f$, we define the corresponding *risk* under $P$: $\mathbb{E}_P R(f) = \mathbb{E}_{(u,i) \sim P} \ell\big(f(u, i), y_{ui}\big)$, the *weighted risk* under $P_w$: $\mathbb{E}_{P_w} R(f) = \mathbb{E}_{(u,i) \sim P} \ell\big(f(u, i), y_{ui}\big) \cdot w(u, i)$.

Our setting covers a wide range of recommendation problems from classical collaborative filtering to sequential recommendation (when user embedding is a function of the item sequence), including those using bandits for exploration where $f(u, i)$ becomes the exploration probability, and the *base measure* is given by the inverse propensities. We defer the detail of that setting to Appendix A.1.

**Consequence of Insufficient Overlapping**.

We now rigorously justify our previous intuition that insufficient overlapping causes uncertainty that fundamentally limits what IW- and DA-based learning can achieve. While the results we show for IW are universal, i.e., it applies to any hypothesis class, we discuss particularly for DA the *skip-gram negative sampling* (SGNS) algorithm (Mikolov et al., 2013). The reason is that SGNS is a widespread technique for learning user or item embedding in IR systems (Xu et al., 2021a), and its more straightforward formulation will allow us to present concise theoretical results to the readers.

As a gentle introduction, IW-based learning uses the ratio $Q(u,i)/P(u,i)$ – where $P$ and $Q$ represent the source and target domain – to reweigh the loss of each instance in order to correct the distribution shift. On the other hand, DA-based learning assumes there exists some hypothesis that makes the *optimal joint risk* of $\inf_{f \in \mathcal{F}}[\mathbb{E}_Q R(f) + \mathbb{E}_P R(f)]$ small. For IW, we show the hardness of learning, i.e., the best target risk $\inf_f \mathbb{E}_Q R(f)$ can be uncontrolled for any $\mathcal{F}$ due to insufficient overlapping. For DA, we show the impossibility of learning, i.e., there does not exist a solution of SGNS that can make the *optimal joint risk* small when $P$ and $Q$ are not aligned.

**Proposition 1.** *Let $P$ and $Q$ be the training and target distribution supported on $\mathcal{S}_P, \mathcal{S}_Q \subseteq \mathcal{D}$, and $w(u,i) = \frac{Q(u,i)}{P(u,i)}$, for $(u,i) \in \mathcal{S}_P \cap \mathcal{S}_Q$. $\mathcal{D}_n$ consists of training instances sampled i.i.d from $P$.*

***Hardness of learning with IW***: *if $\mathcal{S}_P = \mathcal{S}_Q$, for any $f \in \mathcal{F}$, with probability at least $1 - \delta$, it holds:*

$$\mathbb{E}_Q R(f) \lesssim \mathbb{E}_{P_{n,w}} R(f) + \frac{M\left(\log \frac{1}{\delta} + \log \mathcal{N}_\infty(\frac{1}{n}, \mathcal{F})\right)}{n} + \sqrt{\frac{M d_1(P\|Q)\left(\log \frac{1}{\delta} + \log \mathcal{N}_\infty(\frac{1}{n}, \mathcal{F})\right)}{n}},$$

*and $\mathbb{P}\left(\left|\mathbb{E}_Q R(f) - \mathbb{E}_{P_{n,w}} R(f)\right| \gtrsim \sqrt{(d_1(P\|Q) - 1)/n}\right) > 0$ under mild condition, where $M := \max\{w(u,i)\}$, $\mathcal{N}_\infty(\frac{1}{n}, \mathcal{F}) := \mathcal{N}(\frac{1}{n}, \mathcal{F}, \ell_\infty^{2n})$ is the $\frac{1}{n}$-covering number for $\mathcal{F}$ in $\|\cdot\|_\infty$ based on $2n$ i.i.d samples from $P$, and $d_1(P\|Q) = \int_{\mathcal{S}_Q}(dP/dQ)dP$ is a divergence measure. Further, if $\mathcal{S}_P \neq \mathcal{S}_Q$, for any prior and posterior distributions $\pi$ and $\tau$ on $\mathcal{F}$, for risks associated with the 0-1 loss, it holds with probability at least $1 - \delta$:*

$$\mathbb{E}_{f \sim \tau, Q} R(f) \lesssim \mathbb{E}_{f \sim \tau, P_{n,w}} R(f) + \frac{d_{KL}(\pi\|\tau) + \log \frac{d_1(P\|Q)}{\delta}}{n} + \mathbb{E}_{f \sim \tau, \mathcal{S}_Q \setminus \mathcal{S}_P} R(f) + \mathcal{O}(D_n(\tau)),$$

*where $d_{KL}$ is the Kullback-Leibler divergence, $d_1(P\|Q)$ is the same as above but defined only on the overlapped part, and $D_n(\tau)$ consists of disagreement terms detailed in Appendix A.2.*

***Impossibility of learning for DA with SGNS***: *let $R(f)$ be the associated risk for the SGNS algorithm, and $C_{u,i}(P)$, $C_{u,i}(Q)$ be the co-occurrence statistics under $P$, $Q$, as mentioned in Xu et al. (2021a). Even if $\mathcal{S}_P = \mathcal{S}_Q$, it holds that:*

$$\inf_f[\mathbb{E}_P R(f) + \mathbb{E}_Q R(f)] \geq \frac{1}{|\mathcal{S}_P|} \sum_{(u,i) \in \mathcal{S}_P} d_{KL}\big(C_{u,i}(P)\|C_{u,i}(Q)\big) + c,$$

*where $c$ takes some positive value and equals to $\inf_f \mathbb{E}_P R(f) + \inf_f \mathbb{E}_Q R(f)$ when $P = Q$ a.s.*

We defer all the proofs to Appendix A.2. For IW, we reveal the generalization bounds for both the frequentist setting (the first set of bounds) and the PAC-Bayesian setting (the second bound). While they emphasize conceptually on different learning guarantees (Germain et al., 2009), and while the latter can handle the existence of non-overlap ($\mathcal{S}_P \neq \mathcal{S}_Q$), we find them both showing insufficient overlapping (characterized by the discrepancy terms $d_1$ and $d_{KL}$) can cause arbitrarily large gaps leading to poor generalizations to the target domain. For DA with SGNS, our lower bound reveals the impact of insufficient overlapping indirectly via a notion of co-occurrence statistics defined in Xu et al. (2021a). In particular, if the source and target domains are well-aligned in terms of the co-occurrence statistics, the increased divergence term can make it impossible to find a hypothesis that simultaneously performs well in both domains. Although this result holds only for SGNS, it nevertheless suggests the limitations of DA caused by insufficient overlapping in IR.

## 4 MODEL AND THEORY

As much as we would like to continue investigating the consequence of insufficient overlapping, for the purpose of this paper, we have shown the drawbacks of directly applying such as IW and

DA to optimizing recommendation without accounting for the working mechanism of IR systems. In what follows, we describe a principled solution motivated by our domain transportation view, which requires actively identifying the suitable training domain via reweighting. Notably, we no longer perceive reweighting as a change-of-measure tool commonly implied in the causal inference and missing data solutions. Instead, we use reweighing as a mechanism to help identify a particular training domain to ensure certain transportation properties (which we explain next) to the deployment domain[2], which may concern only a subpopulation of all the potential user-item instances.

## 4.1 TRANSPORTATION-CONSTRAINED RISK MINIMIZATION

Throughout our discussion, we emphasize the transportability of patterns across domains, which can be difficult to define precisely. However, if we make a a mild assumption that all patterns can be recovered by Lipschitz-bound functions, it immediately becomes clear that the *Wasserstein distance* can characterize our notion of transportability. To show this point, we first describe an imbalanced binary classification problem with two domains $P$ and $Q$. Let $\ell_1(u) = \frac{u}{p_1}$ and $\ell_{-1}(u) = \frac{u}{1-p_1}$ be the positive and negative loss under prediction score $u$, where $p_1$ is the prior probability that a sample comes from $P$. Interestingly, the $L$-Wasserstein distance between $P$ and $Q$ admits[3]:

$$- d_W(P,Q) = \inf_{g:\{\|g\|_L \leq L\}} p_1 \int_{\mathcal{D}} \ell_1(g)dP + (1-p_1) \int_{\mathcal{D}} \ell_{-1}(g)dQ, \tag{1}$$

which is exactly the optimal risk of the binary classification problem described in the first place! Therefore, Wasserstein distance can be perceived as how well the classification patterns on $P$ transports to $Q$ by any means necessary, as long as they are Lipschitz-bounded. Since the constant of $L$ is simply a scaling factor, we simply continue with the 1-Wasserstein distance. Using $d_W$ to device the domain transportation contraint between $P_f$ and the particular training domain $P_w$ we seek to construct with our weighting mechanism, the learning objective is directly given by:

$$\underset{f \in \mathcal{F}, w \in \mathcal{H}}{\text{minimize}} \ \mathbb{E}_{P_w} R(f) \ \text{ s.t. } D(w,f) \leq \rho, \tag{2}$$

where $D(w, f)$ is a shorthand for $d_W(P_w, P_f)$, and $\rho$ can be treated as a hyper-parameter. As a sanity check, when $\rho = 0$, the feasible set is given by: $f(u, i) = w(u, i)$ for all $(u, i)$, so we are directly optimizing the associated risk of the deployment domain. However, this is a suboptimal practice for deterministic recommenders because in that case, only a small proportion of the feedback data will be considered and the result can be heavily biased toward the historical recommendation. Therefore, a suitable $\rho$ will give us the opportunity to better explore the whole feedback data. Nevertheless, if $\rho$ becomes too large, we are likely to overfit the feedback data due to the high capacity of $f$ and $w$ combined while the transportation constraint is too loose.

The proposed objective is tailored specifically to the working mechanism of IR systems, and it is natural to question the validity and effectiveness. In what follows, we rigorously prove a series of guarantees associated with our novel learning objective, and discuss how it connect and compare to the existing results.

To begin with, we show that jointly optimizing $w$ leads to the sample-consistency guarantee. For clarity, we start with the feature-based learning where $f(u, i) := f(\mathbf{x}_u, \mathbf{x}_i)$, and defer the extension with the representation mapping to the next section. Let $\mathcal{Z} := \mathcal{D} \times \{0, 1\}$, and we denote by $\ell_f : \mathcal{Z} \to \mathbb{R}^+$ the composition with the loss function such that for $\mathbf{z}_{ui} = (\mathbf{x}_u, \mathbf{x}_i, y_{ui})$, we have $\ell_f(u, i) := \ell\big(f(u, i), y_{ui}\big)$. We use $w_{ui}$ as a shorthand for the individual weight.

**Theorem 1.** *Denote by $D_n(w, f) = d_W(P_{n,w}, P_{n,f})$. Suppose that $\ell_f$ is L-Lipschitz w.r.t $\| \cdot \|_2$. Define $S_G = \sup_{g:\|g\|_L \leq 1} \sum_{(u,i)\in\mathcal{D}_n} g(u, i)$ and $M_G = \sup_{\|g\|_L \leq 1, (u,i)\in\mathcal{D}} g(u, i)$. Assume $\mathcal{Z} \in \mathbb{R}^{d_Z}$ and $w \in \mathcal{H}$. Given any $f \in \mathcal{F}$ and $\rho > 0$, it holds with probability at least $1 - \delta$ that:*

$$\left| \min_{w: D(w,f)\leq\rho} \mathbb{E}_{P_w} R(f) - \min_{w: D_n(w,f)\leq\rho} \mathbb{E}_{P_{n,w}} R(f) \right| \leq \frac{9M(\log \frac{1}{2\delta})}{n} + B_n(\mathcal{H}, \delta)L + C_n(\bar{w}, \tilde{w}, f, \delta).$$

---

[2]Suppose a candidate recommender $f$ is deployed to $\mathcal{D}$. According to Section 3, the domain of users and items who are affected by the recommendation is given by $P_f$ defined on $\{(u, i) \in \mathcal{D} : \text{reco}(u, i; f) = 1\}$. We refer to it as the *deployment domain* to differentiate from the notion of *target domain* which is conventionally intervention-free.

[3]In Appendix A.3, we provide a more thorough introduction to Wasserstein distance for interested readers.

*Here,* $\tilde{w} = \arg\min_{D_n(w,f) \leq \delta - B_n(\mathcal{H},\delta)} \mathbb{E}_{P_w} R(f)$, $\bar{w} = \arg\min_{D_n(w,f) \leq \delta + B_n(\mathcal{H},\delta)} \mathbb{E}_{P_w} R(f)$, $C_n(\bar{w}, \tilde{w}, f, \delta) = n^{-\frac{1}{2}} \sqrt{18M \max\{R_{\bar{w}}(f)^2, R_{\tilde{w}}(f)^2\} \log \frac{1}{2\delta}}$ *and* $R_w(f)^2 = \frac{1}{n} \sum_{\mathcal{D}_n} w_{ui}^2 \ell_f(u,i)^2$. *Also,* $B_n(\mathcal{H}, \delta)$ *is given by* $\mathcal{O}\left( \left(S_G n^{1/(d_z+1)}\right)^{-1} + M_G n^{-1/2} \sqrt{\log \frac{1}{2\delta} + \mathcal{N}_{\infty}(\frac{1}{n}, \mathcal{H})} \right)$.

We relegate the proof of this theorem to Appendix A.4. Crucially, other than showing that the sample-consistency result can still be achieved with the jointly-optimized weights, we point out that this bound is completely independent of distribution discrepancy terms that troubled IW and DA. Notice that the bound depends on the weights only through $R_{\bar{w}}(f)^2$ and $R_{\tilde{w}}(f)^2$ presented by the $C_n$ term. Given any $f \in \mathcal{F}$, our result shows that both $\bar{w}$ and $\tilde{w}$ are the minimizers of a weighted risk term, i.e. $\mathbb{E}_{P_w} R(f) \propto \sum_{\mathcal{D}} w_{ui} \ell_f(u,i)$, under some constraint involving $\rho$. Therefore, thanks to the constraint of $D(w,f) \leq \rho$ and the joint optimization of $w$, $R_{\bar{w}}(f)^2$ and $R_{\tilde{w}}(f)^2$ will have more controlled behaviors compared with the domain discrepancy terms in Proposition 1.

We now investigate the learning-theoretical guarantees of the proposed objective, particularly the generalization and excessive risk bounds. Suppose the underlying labeling function of $y_{ui}$ with respect to $(\mathbf{x}_u, \mathbf{x}_i)$ is deterministic. With a little abuse of notation, we use $\mathbf{z} \sim P_{n,f}$ to denote sampling $(\mathbf{x}_u, \mathbf{x}_i, y_{ui})$ via $(\mathbf{x}_u, \mathbf{x}_i) \sim P_{n,f}$ and the deterministic labelling function.

**Theorem 2.** *Suppose* $\inf_{\tilde{\mathbf{z}} \in \mathcal{Z}} \{\ell_f(\tilde{z}) + \lambda \|\mathbf{z} - \tilde{\mathbf{z}}\|_2\} \leq M_{\mathcal{F}}$ *for all* $\mathbf{z} \in \mathcal{Z}$ *and* $\lambda \geq 0, f \in \mathcal{F}$. *Let* $\mathcal{R}_n(\mathcal{F})$ *be the empirical Rademacher complexity:* $\mathcal{R}_n(\mathcal{F}) = \mathbb{E}_{\sigma} \sup_{f \in \mathcal{F}} \left| \frac{1}{n} \sum_{(u,i) \in \mathcal{D}_n} \sigma_{u,i} f(u,i) \right|$, *and* $\sigma$ *takes* $\{-1, 1\}$ *with equal probability. We consider individual weights* $w_{ui} \in (0,1)$ *for brevity.*

*Generalization Error: for any* $f \in \mathcal{F}$, *it holds with probability at least* $1 - \delta$ *that:*

$$\min_{w: D(w,f) \leq \rho} \mathbb{E}_{P_w} R(f) \leq \min_{\lambda \geq 0} \left\{ -\lambda\rho + \mathbb{E}_{\mathbf{z} \sim P_{n,f}} \left[ \inf_{\tilde{\mathbf{z}} \in \mathcal{Z}} \ell_f(\tilde{z}) + \lambda \|\mathbf{z} - \tilde{\mathbf{z}}\|_2 \right] + M_{\mathcal{F}} \sqrt{\frac{\log 1/(2\delta)}{n}} \right.$$
$$\left. + \sqrt{\frac{\log(\lambda+1)}{n}} + \frac{C \int_0^{\infty} \sqrt{\log \mathcal{N}_{\infty}(\epsilon, \mathcal{F})} d\epsilon}{n^{1/2}} \right\}.$$

*Excessive risk: define* $\hat{f} = \arg\min_{f, D_n(w,f) \leq \rho} \mathbb{E}_{P_{n,w}} R(f)$ *as the empirical optimal and* $f^* = \arg\min_{f, D(w,f) \leq \rho} \mathbb{E}_{P_w} R(f)$ *as the theoretical optimal. With probability at least* $1 - \delta$:

$$\left| \min_{w: D(w,\hat{f}) \leq \rho} \mathbb{E}_{P_w} R(\hat{f}) - \min_{w: D(w,f^*) \leq \rho} \mathbb{E}_{P_w} R(f^*) \right| \leq 2\mathcal{R}_n(\ell \circ \mathcal{F}) + \mathcal{O}\left( \frac{L + \tilde{M}\sqrt{\log 1/\delta}}{n^{1/2}} \right).$$

We defer both proofs to Appendix A.5. Theorem 2 establishes two critical learning guarantees for the proposed transportation-constraint risk minimization. In particular, the generalization error reveals how we resolve the previous drawback mentioned for using DA in IR system. Recall from Section 3 that the *joint optimal risk* might be uncontrolled for DA since it can depend on how different the training and target domains are. In this regard, with $D(w,f) \leq \rho$ explicitly constrains the discrepancy between the training and deployment domain, we have the additional $-\lambda\rho$ term in the generalization error bound where the second term on the RHS can be think of as the empirical "transported" risk. In other words, the constraint ensures that the deployment risk will not deviate far from the training risk, and thus creates the possibility of learning transportable recommendation. Further, the excessive risk bound suggests that the transportation constraint does not cost the convergence rate comparing with the standard empirical risk minimization guarantee (Mohri et al., 2018).

### 4.2 TOWARDS TRANSPORTATION-REGULARIZED RISK MINIMIZATION

To render the proposed objective computationally feasible, we apply a strong duality argument to show that there exists a regularized counterpart for (2), where we defer the proof to Appendix A.6.

**Claim 1.** *Suppose that* $\ell \circ \mathcal{F}$ *is convex. It holds for any* $f \in \mathcal{F}$ *that:*

$$\min_{P_w: d_W(P_w, P_f) \leq \rho} \mathbb{E}_{P_w} R(f) = \max_{\lambda \geq 0} \left\{ -\lambda\rho + \min_{P_w} \left\{ \mathbb{E}_{P_w} R(f) + \lambda \cdot d_W(P_w, P_f) \right\} \right\}.$$

The above duality result suggests we alternatively consider the regularized learning objective:

$$\underset{f \in \mathcal{F}, w \in \mathcal{H}}{\text{minimize}} \, \mathbb{E}_{P_w} R(f) + \lambda \cdot d_W(P_w, P_f), \tag{3}$$

where the relaxation on $\max_{\lambda \geq 0}$ has been made. Here, $\lambda$ is the tuning parameter that regularizes the domain transportation between the reweighted training domain and the deployment domain. To give some insight for the transportation-regularized objective, we use learning from bandit feedback data as a test bed and examine how the optimal solution of (3) compares to the solution of the standard counterfactual risk minimization with complete overlapping (Swaminathan and Joachims, 2015a). Let the *logging policy* by given by $\pi$. It holds in the bandit off-policy learning setting that: $P_f(u, i) \propto \frac{f(u,i)}{\pi(u,i)}$, where $f(u, i) \in (0, 1)$ now serves as the policy we try to optimize. The *counterfactual risk* is given by: $\mathbb{E}_{P_f} R(f) := \sum_{(u,i) \in \mathcal{D}} \frac{f(u,i)}{\pi(u,i)} \ell(f(u, i), y_{ui})$, and we prove that the solution of (3) is consistent with that of the counterfactual risk minimization.

**Theorem 3.** *Let $f^* = \arg \min_{f \in \mathcal{F}} \mathbb{E}_{P_f} R(f)$ be the solution of the counterfactual risk minimization, and $\hat{f}, \hat{w}$ be the solution of the proposed objective in (3). For suitably large $\lambda$ s.t. $\lambda \geq \mathcal{O}(L) \geq 1$, it holds with probability at least $1 - \delta$ that:*

$$\left| \mathbb{E}_{P_{f^*}} R(f^*) - \left( \mathbb{E}_{P_{\hat{w}}} R(\hat{f}) + \lambda d_W(P_{\hat{w}}, P_{\hat{f}}) \right) \right| \lesssim \frac{\lambda}{n^{1/(d+1)}} + \frac{\log 1/(2\delta)}{n} + \lambda M_G \sqrt{\frac{\log 1/(2\delta)}{n}}$$

$$+ \sqrt{\frac{M \max\{R_{f^*/\pi}(f^*)^2, R_{\tilde{w}}(\tilde{f})^2\} \log \frac{1}{2\delta}}{n}},$$

*where $M_G$, $R_w(f)^2$ are defined as in Theorem 1, and $\pi$ is the logging policy.*

Intuitively, since both $f$ and $\pi$ are probability distributions with the same support in the idea bandit feedback setting, a suitably large $\lambda$ will drive $P_w$ ultimately toward $P_f$ and lead to the consistency result. We defer the proof of Theorem 3 to Appendix A.7. This example reveals the effectiveness of the proposed transportation-regularized risk minimization, which we show is a strict extension of the counterfactual risk minimization. For IR systems, the transportation-regularized risk minimization applies to a much broader setting including deterministic recommendation. Nonetheless, directly solving (3) is still challenging due to the complications in computing the Wasserstein distance. Inspired by how Wasserstein GAN transforms its objective to a two-player adversarial game (Gulrajani et al., 2017), we can also convert our objective to a minimax optimization problem discussed below.

## 4.3 MINIMAX OPTIMIZATION

---

**Algorithm 1:** Batch-wise GDA for Minimax Optimization

---

**Input:** $\tilde{\mathcal{D}}(t)$: a batch of training sample of size $m$ at step $t$. $\eta$: the learning rate for ascent step. $\gamma$: shrinking parameter for the learning rate of descent step. $\lambda$: the regularization parameter.

**for** *each step $t = 0, 1, \ldots$* **do**

    Compute $\text{reco}(u, i; f_{\theta_1})$ for $(u, i) \in \tilde{\mathcal{D}}(t)$ (see Section 5 for the computation detail);

    $\theta_1^{(t+1)} \leftarrow \theta_1^{(t)} - \frac{\eta}{\gamma} \nabla_{\theta_1} \mathcal{L}_\lambda([\theta_1^{(t)}, \theta_2^{(t)}], \theta_3^{(t)}); \theta_2^{(t+1)} \leftarrow \theta_2^{(t)} - \frac{\eta}{\gamma} \nabla_{\theta_2} \mathcal{L}_\gamma([\theta_1^{(t)}, \theta_2^{(t)}], \theta_3^{(t)});$

    $\theta_3^{(t+1)} \leftarrow \theta_3^{(t)} + \eta \nabla_{\theta_3} \mathcal{L}_\lambda([\theta_1^{(t)}, \theta_2^{(t)}], \theta_3^{(t)}).$

**end**

---

The key to our implementation is again perceiving Wasserstein distance as a binary classification problem with Lipschitz-bounded classifier as we discussed in Section 4.1. Towards this end, we denote by $g \in \mathcal{G}$ any suitable classifier for computing $d_W$ according to (1). Without loss of generality, we let $w$, $f$ and $g$ be parameterized by $\theta_1$, $\theta_2$ and $\theta_3$. It is straightforward to verify that the regularized objective in (3) transforms exactly into: $\min_{[\theta_1, \theta_2]} \max_{\theta_3} \mathcal{L}_\lambda([\theta_1, \theta_2], \theta_3)$, where $\mathcal{L}_\lambda$ given by:

$$\sum_{(u,i) \in \mathcal{D}} w_{ui}(\theta_2) \ell_{ui}(f_{\theta_1}) + \lambda \Big( \sum_{(u,i) \in \mathcal{D}} w_{ui}(\theta_2) \cdot g_{ui}(\theta_3) - \sum_{(u,i) \in \mathcal{D}} \text{reco}(u, i; f_{\theta_1}) \cdot g_{ui}(\theta_3) \Big). \tag{4}$$

In view of (4) as a two-player game, the first player optimizes $f$ and $w$ to minimize both the training risk and domain transportation, while the second player tries to increase the gap between the two domains. The global equilibrium is reached when neither player can change the objective without knowing each other's strategy, which means $f$ and $w$ cannot simultaneously improve the empirical risk and the domain transportation between $P_w$ and $P_f$. It suggests we have found the recommendation hypothesis that best transports the learnt patterns to its intervention domain. We propose using the two-time-scale gradient descent-ascent algorithm (GDA) to solve the minimax optimization.

GDA is very efficient for solving objectives like (4) with both empirical and theoretical evidence (Lin et al., 2020), and it has been widely applied to optimize generative adversarial networks. Since our solution will be deployed to real-world systems, we pay more attention to the convergence and stability of GDA. In appendix A.8, we rigorously prove that for the bounded Lipschitz functions we consider, GDA will converge to the global Nash equilibrium with a proper learning-rate schedule. Further, it is unlikely to bouncing around near the optimum. Now that we have the performance guarantee, we apply the batch-wise GDA described in Algorithm 1 to solve our objective.

## 5 EXPERIMENT AND RESULT

We present in this section the experiment settings, implementations, empirical results, as well as the analysis. The complete experimental results and the real-world online testing performance are deferred to Appendix C. All the implementation codes are provided in the supplement material.

**Computing the reco$(u, i; f)$ function**. For the deterministic top-K recommendation, reco$(u, i; f)$ requires computing and sorting the scores of $\{f(u, i_1), \ldots, f(u, i_{|\mathcal{I}|})\}$ for the user $u$. This is expensive during training, so we refer to the widespread practice of downsampling. In particular, for each $(u', i')$, we randomly sample $m < |\mathcal{I}|$ *irrelevant* items and see if $i'$ is among the top-K (Krichene and Rendle, 2020). While the sampled outcome may have certain bias, it significantly speeds up the training process while maintaining a decent accuracy under moderate $m$, e.g. $m = 100$.

**Benchmark datasets.** We refer to the *MovieLens-1M*, *LastFM*, and *GoodReads* datasets that are extensively employed by the IR literature. We treat them as implicit feedback data, and the detailed description and processing are deferred to Appendix C.1.

**Simulation studies.**. For implicit feedback, the response (click) is generated from the exposure and relevance status via: $\mathbb{P}(Y_{ui} = 1) = \mathbb{P}(O_{ui} = 1) \cdot \mathbb{P}(R_{ui} = 1)$, where $O_{ui}$ indicates the exposure and $R_{ui}$ is the relevance. The relevance score is critical for unbiased offline evaluation, but it is unknown for real-world datasets. We thus conduct simulation so we know what the ground truth relevance is. To make sure the distribution of the simulated data conforms to the real-world feedback data, we first extract the exposure and relevance signals from public datasets (e.g. the three datasets described above) via models such as matrix factorization, and then generate the new feedback data by adding noise to the extracted relevance signal. The detailed mechanisms are provided in Appendix C.1.1.

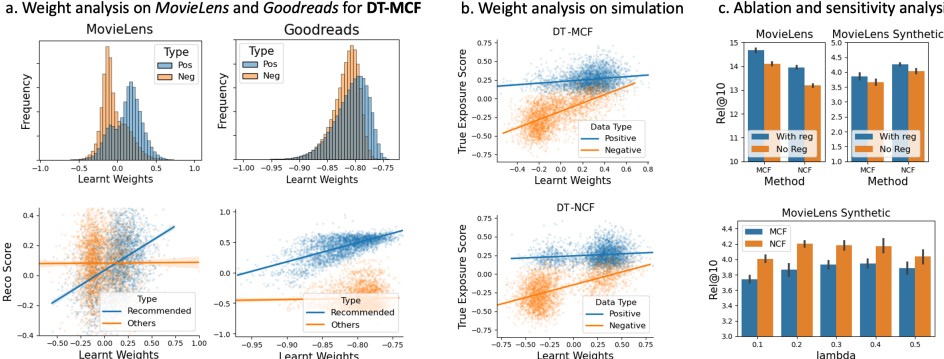

Figure 2: (a). Weight analysis on *MovieLens* and *GoodReads*. Upper: distribution of $w_{ui}$ on the positive and negative feedback data. Lower: relation between $w_{ui}$ and $f_{ui}$ under $P_f$ (the recommended $(u, i)$ under $f$) and others; (b). Weight analysis on *MovieLens* simulation: the relations between $w_{ui}$ and $\mathbb{P}(O_{ui} = 1)$ on the positive and negative synthetic feedback data for MCF (upper) and NCF (lower); (c). Ablation study for the transportation regularization and sensitivity analysis on $\lambda$ for *MovieLens* and its synthetic data.

Table 1: Real-data and semi-synthetic experiment *testing* testing results for the MovieLens-1M, LaftFM and GoodReads dataset. All the results are computed via ten runs and the Hit and NDCG metrics are multiplied by 100 for presentation. The results for the *LastFM* and *GoodReads* datasets, and sequential recommendation (*Attn*) are deferred to Appendix D.3. The metric of **Rel@10**, **Hit@10** and **NDCG@10** are short for the top-10 relevance score, hitting rate and normalized discounted cumulative gain. The metrics on synthetic data are computed by using the true relevance model. We highlight the **best** and second best results. **Attn** is not experimented on simulation because the ordering of the generated data is random.

|  | Pop | IPW-MF | ExpoCF | AC-MF | MCF | DT-MCF | NCF | DT-NCF | Attn | DT-Attn |
|---|---|---|---|---|---|---|---|---|---|---|
| **MovieLens-1M** | | | | | | | | | | |
| Rel@10 | 9.05 (.02) | 14.00 (.05) | 14.04 (.08) | 14.05 (.07) | 14.02 (.03) | 14.65 (.07) | 13.77 (.04) | 13.98 (.06) | 14.26 (.10) | **16.32** (.09) |
| Hit@10 | 43.67 (.10) | 60.48 (.12) | 60.21 (.14) | 61.19 (.11) | 60.27 (.15) | 61.95 (.12) | 59.95 (.13) | 61.17 (.15) | 72.23 (.19) | **74.82** (.24) |
| NDCG@10 | 22.73 (.03) | 31.17 (.09) | 30.98 (.10) | 31.72 (.11) | 30.91 (.11) | 32.83 (.08) | 31.07 (.10) | 32.27 (.07) | 42.48 (.10) | **43.09** (.14) |
| **MovieLens-1M Simulation** | | | | | | | | | | |
| Rel@10* | 2.85 (.02) | 3.77 (.02) | 3.73 (.03) | 3.79 (.03) | 3.75 (.02) | 3.98 (.02) | 3.96 (.02) | **4.11** (.03) | - | - |
| Hit@10 | 56.12 (.07) | 75.02 (.11) | 74.87 (.09) | 74.98 (.13) | 75.64 (.09) | 76.66 (.12) | 75.69 (.09) | **76.87** (.11) | - | - |
| NDCG@10 | 29.71 (.05) | 37.79 (.06) | 37.86 (.07) | 37.77 (.07) | 37.70 (.05) | 38.57 (.08) | 38.43 (.05) | **38.72** (.07) | - | - |
| **LastFM** | | | | | | | | | | |
| Rel@10 | 1.63 (.01) | 6.41 (.04) | 6.44 (.03) | 6.48 (.03) | 6.38 (.03) | **6.59** (.02) | 6.20 (.05) | 6.56 (.04) | 5.71 (.04) | 5.82 (.05) |
| Hit@10 | 25.22 (.01) | 79.21 (.31) | 79.62 (.27) | 79.64 (.22) | 79.02 (.22) | **81.81** (20) | 77.20 (.35) | 80.83 (.33) | 67.48 (.35) | 68.74 (.37) |
| NDCG@10 | 15.35 (.01) | 50.63 (.19) | 50.76 (.22) | 51.05 (.19) | 50.06 (.13) | 52.96 (.14) | 50.55 (.18) | 52.03 (.26) | 56.12 (.22) | **56.60** (.19) |
| **GoodReads** | | | | | | | | | | |
| Rel@10 | 4.98 (.01) | 5.94 (.08) | 5.93 (.10) | 5.97 (.07) | 5.90 (.08) | **6.14** (.07) | 5.84 (.02) | 6.07 (.09) | 4.57 (.04) | 4.68 (.10) |
| Hit@10 | 43.37 (.06) | 59.04 (.10) | 59.28 (.14) | 59.33 (.12) | 58.32 (.12) | **60.95** (.11) | 58.07 (.04) | 59.73 (.34) | 48.97 (.26) | 49.30 (.25) |
| NDCG@10 | 26.16 (.03) | 35.19 (.07) | 35.22 (.08) | 35.45 (.11) | 34.69 (.10) | **36.45** (.09) | 34.90 (.02) | 36.27 (.15) | 25.76 (.17) | 26.04 (.13) |

**Models and baselines methods.** The functions of $f$, $w$ and $g$ in the minimax objective (4) can be any recommendation algorithm that suits the context, such as matrix-factorization collaborative filtering (**MCF**), neural collaborative filtering (**NCF**) (He et al., 2017), and sequential recommendation model with the prevalent attention mechanism (**Attn**) as in Kang and McAuley (2018). We index our domain domain transportation approach by such as **DT-X**, with **X** stands for the models of $f$, $w$ and $g$. We will experiment with a broad range of combinations for $f$, $w$ and $g$ as we present in Appendix C.3. As for the baseline methods, other than the standard popularity-based recommendation (**Pop**) and MCF, we select particularly the approaches that adapts causal inference or missing data techniques, including the IPW-debaised MF method (**IPW-MF**) from Saito et al. (2020), the user-exposure aware MF (**ExpoMF**) from Liang et al. (2016), and the adversarial-counterfactual MF method (**AC-MF**) from Xu et al. (2020). The model configurations and training details are deferred to Appendix C.2.

**Training & evaluation**. We use the two-timescale GDA as described in Algorithm 1 with a pre-selected shrinkage parameter, and we will conduct its sensitivity analysis in Appendix C.4. The configurations for both our model and the baselines are described in Appendix C.2. We adopt the convention where the second-to-last interaction is used for validation, and the last interaction is used for testing. For evaluation, in addition to the regular *Hit@10* and *NDCG@10* as in Rendle et al. (2020), we further consider the relevance score of the top-10 recommendation, i.e. $\sum_{\text{top-10}} R_{ui}$, since it also reveals the quality of the recommendation and more closely resembles the deployment performance of $\mathbb{E}_{P_f} R(f)$. We have access to $R_{ui}$ during simulation, and on real-data analysis, we settle to the $Y_{ui}$ in testing data as an approximation. The evaluation results are provided in Table 1.

**Result analysis**. From Table 1, we first observe that the proposed methods achieves the best performances in both the benchmark and simulated datasets on all metrics. In particular, the **DT-X** method significantly improves the performance of the original model **X**. Also, our solution outperforms all the baselines, including those adapted from causal inference or missing data methods. The rest simulation studies are provided in Appendix C.3, and their results admit similar patterns. Finally, we conduct **ablation studies** on the transportation regularizer and the **sensitivity analysis** on $\lambda$ (see Figure 2c). It is clear that without the transportation regularization, the performances degenerate significantly in both the real data and simulation studies. Also, we find that the performance variation of the proposed approach is relatively steady under different $\lambda$.

**Analysis on** $w$. We specifically analyze the learnt weights since they play a critical role in our approach. On both *MovieLens* and *GoodReads* (Figure 2a), we find that: 1). the learnt weights are larger on the positive training instances; 2). for the $(u, i)$ that are among $P_f$ (the recommended), the weights $w_{ui}$ are higher and relates positively with $f(u, i)$, while there is no such pattern for the non-recommended. They both suggest the weights are indeed learnt to balance the domains of the positive feedback data and the instances recommended under $f$. In Figure 2b, we observe from the *Movielens* simulation study that the learnt weights correlate positively to the exposure on the

negative training data, and are distributed more evenly on the positive data. It means that $w_{ui}$ learns to emphasize the instances that were more likely to be exposed but not clicked, which usually have strong negative signals, and the positive instances. It is valuable for training under implicit feedback because the algorithm learns to locate the strong negative instances. The remaining weight analysis, ablation studies, and sensitivty analysis are provided in Appendix C.3.

**Real-world IR system.** We deploy our solution to a real-world IR system held by a major e-commerce platform in the U.S. We deploy our transportation-regularized solution, and the online A/B/C testing against the standard and IPW-based solutions for an item recommendation task. Due to the space limitation, we defer the details and analysis to Appendix C.6.

## 6 CONCLUSION AND DISCUSSION

In this paper, we study from the critical intervention perspective of recommendation and summarize a domain transportation view supported by the realistic working mechanism of IR systems. We propose a principled transportation-constraint risk minimization for optimizing recommendation, and illustrate by both theories and experiments how we find it more favorable than directly importing the causal inference and missing data solutions. However, we point out that our work does not aim to replace those solutions since we focus exclusively on optimizing recommendation. For instance, we do not readily incorporate the known causal mechanisms (which might be available for certain applications) to answer questions that are relevant to the decision making in IR. It will be interesting to explore how the presumed or discovered causal structures can interact with domain transportation and improve our solution. Towards this end, our transportation-constraint risk minimization can be viewed as a modular toolkit that allows researchers to apply further insights to solve more complex IR problems. Finally, we hope our work raises the awareness of developing IR-oriented solutions with a proper understanding of the system's mechanism. The novel perspectives, solutions, and theoretical results in our paper may also open the door to future research.

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

## A APPENDIX ON BACKGROUND AND MAIN THEORETICAL RESULTS

We provide the supplementary background introduction, proofs for the claim, proposition and theorem, as well as the detailed experiment settings and complete numerical results in this part of the paper.

### A.1 RECOMMENDER WITH EXPLORATION POLICY, AND LEARNING FROM BANDIT FEEDBACK DATA

Using multi-armed bandits for exploration has attracted increasing attention from the recommendation literature in recent years. The major advantage is that the interactive process between users and recommender is explicitly characterized in an online fashion. Exploration also helps alleviate the exposure bias, popularity bias, position bias, etc, via more efficient exploitation of the feedback data. A critical difference between the bandit recommender and the deterministic recommender we referred to in our paper is that an item is exposed to the user with randomness, and the exploration policy controls the randomness in the exposure.

Mathematically, we continue with $\mathcal{X}$ as the context (feature) space, $\mathcal{I} = \{1, \ldots, k\}$ be the action (item) space. The exploration-exploitation strategy can be viewed as a sequential game between the user and recommender. For notation simplicity, we introduce the stationary bandit setting where the exploration policy does not depend on the history. In each round, an item is sampled according to the exploration policy $\pi(u, i) := \pi(i \,|\, \mathbf{x}_u)$, which is a mapping between the item space and the probability space, and expose it to the user. Then the user, whose feature is $\mathbf{x}_u$, reveal the regret (or reward) action $i \in \mathcal{I}$, which corresponds to the $Y_{u,i}$ in our setting. In the context of delivering recommendation, clearly we have: $\mathbb{P}(\text{reco}(u, i; f) = 1) = \pi(u, i)$ during exploration, and in the context of learning from the logged feedback data, we have $\mathbb{P}(O_{ui} = 1) = \pi(u, i)$ since it characterizes the probability of exposure.

Similar to the deterministic recommender, the challenge in bandit offline learning is largely due to its partial-observation nature as we do not observe the complete reward for all items. However, unlike the deterministic recommender, the random exposure mechanism allows efficient estimation of the *counterfactual* regret, i.e. what the regret would have become if an alternative exploration policy was used. The IPW estimator fits particularly well to this purpose, since it provides the unbiased estimation of the counterfactual regret:

$$\mathbb{E}_{P_\pi} \frac{f(u, i)}{\pi(u, i)} Y_{ui} = \mathbb{E}_{P_f} Y_{ui}.$$

Swaminathan and Joachims (2015a) refers to the IPW-reweighted risk, which essentially replace the regret $Y_{ui}$ by the corresponding risk under $f$, as the *counterfactual risk* for learning from the logged bandit feedback data. It exactly corresponds to setting the base measure $P$ with $P_{\text{IPW}}$, so it holds: $\mathbb{E}_{P_f} R(f) = \sum_{(u,i) \in \mathcal{D}} \frac{f(u,i)}{\pi(u,i)} \ell(f(u, i), y_{ui})$.

### A.2 PROOF AND BACKGROUND FOR THE IW AND DA BOUNDS IN PROPOSITION 1

We first prove the upper bound for importance weighting under weak overlap .

*Proof.* Our proof leverages the classical "double sampling" technique from Anthony and Bartlett (2009). We use $\vec{z} = [\mathbf{z}_1, \ldots, \mathbf{z}_n]$ to denote the observed samples, and $\vec{z}' = [\mathbf{z}'_1, \ldots, \mathbf{z}'_n]$ to denote an i.i.d copy of $\vec{z}$. We first define by:

$$UB_1(f, \vec{z}, t) = \frac{1}{n} \sum_{i=1}^{n} w_i \ell_f(\mathbf{z}_i) + \frac{3Mt}{n} + \sqrt{\frac{2d(P\|Q)t}{n}},$$

and

$$UB_2(f, \vec{z}, t) = \frac{1}{n} \sum_{i=1}^{n} w_i \ell_f(\mathbf{z}_i) + \frac{9Mt}{n} + \sqrt{\frac{18d(P\|Q)t}{n}}.$$

Given $f \in \mathcal{F}$, let $A := \mathbb{E}_Q R(f) + \frac{6Mt}{n} + \sqrt{\frac{8d(P\|Q)t}{n}}$ it holds:

$$\mathbb{P}\big(UB_2(f, \vec{\mathbf{z}}', t) \leq UB_1(f, \vec{\mathbf{z}}, t)\big) \leq \mathbb{P}\big(UB_2(f, \vec{\mathbf{z}}', t) \leq A\big) + \mathbb{P}\big(UB_1(f, \vec{\mathbf{z}}, t) \geq A\big)$$

$$\leq 2\mathbb{P}\Big(\big|\mathbb{E}_Q R(f) - \frac{1}{n}\sum w_i \ell_f(\mathbf{z}_i)\big| \geq \frac{3Mt}{n} + \sqrt{\frac{2d(P\|Q)t}{n}}\Big)$$

$$\leq 4e^{-t},$$

where the last line follows from Lemma A.2. Next, we define $\mathcal{C}(\epsilon, \ell \circ \mathcal{F}, \ell_1(P_{n,w}))$ be the $\epsilon$-cover of $\ell \circ \mathcal{F}$ with the empirical $\ell_1$ norm under $P_{n,w}$ such that for any $f \in \ell \circ \mathcal{F}$, there exists $\tilde{f}$ in $\mathcal{C}(\epsilon, \ell \circ \mathcal{F}, \ell_1^n)$: $\big|\frac{1}{n}\sum w_i f(\mathbf{z}_i) - \frac{1}{n}\sum w_i \tilde{f}(\mathbf{z}_i)\big| \leq \epsilon$, for $(\mathbf{z}_1, \ldots, \mathbf{z}_n)$ sampled i.i.d from $P$. It then holds:

$$\mathbb{P}\big(\exists f \in \mathcal{F} : \mathbb{E}_Q R(f) \geq UB_2(f, \vec{\mathbf{z}}, t) + \epsilon\big)$$

$$= \mathbb{E}_{\vec{\mathbf{z}}} \sup_{f \in \mathcal{F}} I[\mathbb{E}_Q R(f) \geq UB_2(f, \vec{\mathbf{z}}, t) + \epsilon]$$

$$\overset{(a)}{\leq} \mathbb{E}_{\vec{\mathbf{z}}} \sup_{f \in \mathcal{F}} I[\mathbb{E}_Q R(f) \geq UB_2(f, \vec{\mathbf{z}}, t) + \epsilon] \cdot 2\mathbb{E}_{\vec{\mathbf{z}}'} I[UB_1(f, \vec{\mathbf{z}}', t) \geq \mathbb{E}_Q R(f)]$$

$$\leq 2\mathbb{E}_{\vec{\mathbf{z}}, \vec{\mathbf{z}}'} \sup_{f \in \mathcal{F}} I[UB_1(f, \vec{\mathbf{z}}', t) \geq UB_2(f, \vec{\mathbf{z}}, t) + \epsilon]$$

$$\overset{(b)}{\leq} 2\mathbb{P}_{\sigma(\vec{\mathbf{z}}, \vec{\mathbf{z}}')}\big(\exists \tilde{f} \in \mathcal{C}(\epsilon, \ell \circ \mathcal{F}, \ell_1(P_{n,w})) : UB_1(f, \sigma(\vec{\mathbf{z}}, \vec{\mathbf{z}}'), t) \geq UB_2(f, \sigma(\vec{\mathbf{z}}, \vec{\mathbf{z}}'), t)\big)$$

$$\leq 8\mathcal{N}(\epsilon, \ell \circ \mathcal{F}, \ell_1(P_{n,w})) \cdot e^{-t},$$

where (a) follows from the fact that $\mathbb{E}_{\vec{\mathbf{z}}'} I[UB_1(f, \vec{\mathbf{z}}', t) \geq \mathbb{E}_Q R(f)] \geq \frac{1}{2}$ as suggested by Lemma A.2, and in step (b) we let $\sigma(\vec{\mathbf{z}}, \vec{\mathbf{z}}')_i$ takes the value of $\mathbf{z}_i, \mathbf{z}'_i$ with equal probability, and the inequality follows from the definition of the $\epsilon$ cover. Notice that $\mathcal{N}(\epsilon, \ell \circ \mathcal{F}, \ell_1(P_{n,w})) \leq \mathcal{N}(\epsilon/M, \ell \circ \mathcal{F}, \ell_\infty^n)$. We take $\epsilon = \frac{1}{n}$, which solves for $t = c \log \frac{1}{\delta} + \log \mathcal{N}(\epsilon/M, \ell \circ \mathcal{F}, \ell_\infty^n)$ for some constant $c$ and leads to the desired result after arranging terms.

The corresponding lower bound follows directly from Lemma A.4, where we make a mild assumption that there exists a hypothesis $f \in \mathcal{F}$ such that $\mathbb{E}_{P_n} R(f) = 1$, so it holds that $\mathbb{E}_{P_{n,w}} R(f) = d(P\|Q) - 1$ suggested by Lemma 1 of Cortes et al. (2010). $\qquad \square$

We then show the PAC-Bayesian result that stated for the partial-overlap case. Our proof relies on the existing results in the PAC-Bayesian literature as in Germain et al. (2015).

*Proof.* We use $\ell$ to denote the $0-1$ loss for brevity. The PAC-Bayesian approach concerns with the Gibbs risk of $\mathbb{E}_{f\sim\tau}\mathbb{E}_{(u,i)\sim Q}\ell(f(u,i), y_{ui})$, where $\tau$ is some posterior distribution on the hypothesis space $\mathcal{F}$. Clearly, when there are mismatch in the supports of $\mathcal{S}_P$ and $\mathcal{S}_Q$, the Gibbs risk for the mismatched part is given by $\mathbb{E}_{f\sim\tau}\mathbb{E}_{(u,i)\in\mathcal{S}_Q/\mathcal{S}_P} R(f)$. We now focus on the overlapped part. Using the classical decomposition result from Germain et al. (2015), which states that:

$$\mathbb{E}_{f\sim\tau}\mathbb{E}_Q R(f) = \frac{1}{2}d_Q(\tau) + e_Q(\tau),$$

where $d_{\mathcal{D}}(\tau) = \mathbb{E}_{f,f'\sim\tau}\mathbb{E}_Q \ell(f(u,i), f'(u,i))$ is a discrepancy term, and $e_Q(\tau) = 2\mathbb{E}_{f,f'\sim\tau}\mathbb{E}_Q \ell(f(u,i), y_{ui})\ell(f'(u,i), y_{ui})$ is the Gibbs risk for the joint hypothesis. Therefore, we have:

$$\mathbb{E}_{f\sim\tau}\mathbb{E}_Q R(f) - \mathbb{E}_{f\sim\tau}\mathbb{E}_{P_w} R(f) = \big(\frac{1}{2}d_Q(\tau) - \frac{1}{2}d_{P_w}(\tau)\big) + \big(e_Q(\tau) - e_P(\tau)\big)$$

$$\leq \frac{1}{2}\underbrace{\big|d_Q(\tau) - d_{P_w}(\tau)\big| + \big|e_Q(\tau) - e_{P_w}(\tau)\big|}_{D(\tau)},$$

where $D(\tau)$ essentially characterizes the disagreement of $Q$ and $P_w$ in terms of the posterior $\tau$ on $\mathcal{F}$. Therefore, for any posterior distribution $\tau$, we have: $\mathbb{E}_{f\sim\tau}\mathbb{E}_Q R(f) \leq \mathbb{E}_{f\sim\tau}\mathbb{E}_{P_w} R(f) + D(\tau),$

so the next step is to bound the Gibbs risk of $\mathbb{E}_{f\sim\tau}\mathbb{E}_{P_w}R(f)$. Our strategy follows that of Germain et al. (2015), that we employ the convex function $\Delta : [0, M] \times [0, M] \to \mathbb{R}$. It then holds that:

$$
n \cdot \Delta\Big(\mathbb{E}_{f\sim\tau}\mathbb{E}_{P_{n,w}}R(f), \mathbb{E}_{f\sim\tau}\mathbb{E}_{P_w}R(f)\Big)
$$

$$
\leq n\mathbb{E}_{f\sim\tau}\Delta\Big(\mathbb{E}_{P_{n,w}}R(f), \mathbb{E}_{P_w}R(f)\Big)
$$

$$
\leq d_{KL}(\tau\|\pi) + \ln\mathbb{E}_{f\sim\pi}\exp\Big(n\Delta\Big(\mathbb{E}_{P_{n,w}}R(f), \mathbb{E}_{P_w}R(f)\Big)\Big) \quad \text{by Lemma A.7}
$$

$$
\overset{\text{w.p. }1-\delta}{\leq} d_{KL}(\tau\|\pi) + \ln\frac{1}{\delta}\mathbb{E}_{f\sim\pi}\mathbb{E}_{\mathbf{z}_1,\dots,\mathbf{z}_n\sim P}\exp\Big(n\Delta\Big(\frac{1}{n}\sum_k w(\mathbf{z}_k)\ell_f(\mathbf{z}_k), \mathbb{E}_{\mathbf{z}\sim P}w(\mathbf{z})\ell_f(\mathbf{z})\Big)\Big),
$$

where the last line comes from the Markov inequality, and $w(\mathbf{z}_k)$ is a short-hand for $w(u, i)$ for $\mathbf{z}_k = (u, i, y_{ui})$. We then focus on bounding: $\mathbb{E}_{f\sim\pi}\mathbb{E}_{\mathbf{z}_k\sim P}\exp\Big(n\Delta\Big(\frac{1}{n}\sum_k w(\mathbf{z}_k)\ell_f(\mathbf{z}_k), \mathbb{E}_{\mathbf{z}\sim P}w(\mathbf{z})\ell_f(\mathbf{z})\Big)\Big)$. Since $\ell_f$ is the $0-1$ loss, the term inside $\Delta$ (setting aside $w(\mathbf{z}_k)$) are essentially binomial variable of $n$ trials with a success rate of $\mathbb{E}_P R(f)$. Since $\Delta$ is convex in both arguments and $w(u, i)$ is given by $Q(u, i)/P(u, i)$, it is easy to verify that:

$$
\mathbb{E}_{\mathbf{z}_k\sim P}\exp\Big(n\Delta\Big(\frac{1}{n}\sum_k w(\mathbf{z}_k)\ell_f(\mathbf{z}_k), \mathbb{E}_{\mathbf{z}\sim P}w(\mathbf{z})\ell_f(\mathbf{z})\Big)\Big)
$$

$$
\leq d(P, Q) \cdot \mathbb{E}_{\mathbf{z}_k\sim P}\exp\Big(n\Delta\Big(\frac{1}{n}\sum_k \ell_f(\mathbf{z}_k), \mathbb{E}_P R(f)\Big)\Big),
$$

where $d(P, Q) = \int_{\mathcal{S}_Q}(dP/dQ)dP$ as we mentioned in the statement. It is then obvious that:

$$
\mathbb{E}_{\mathbf{z}_k\sim P}\exp\Big(n\Delta\Big(\frac{1}{n}\sum_k \ell_f(\mathbf{z}_k), \mathbb{E}_P R(f)\Big)\Big) \leq \sup_{r\in[0,1]}\Big\{\sum_{k=0}^n \text{Bin}(k; n; r)\exp\big(n\Delta(\frac{k}{n}, r)\big)\Big\},
$$

where we use $r \in [0, 1]$ to replace the role of $\mathbb{E}_P R(f)$, since the prior $\pi$ is arbitrary, and $\text{Bin}(k; n; r)$ is probability of having $k$ success in $n$ rounds of Bernoulli trail where the success rate is $r$. It is shown by Germain et al. (2009) that the RHS is $\mathcal{O}(1)$ by choose a particular $\Delta$, so we obtain the desired result by putting together the above arguments. □

Now we briefly introduce the skip-gram negative sampling algorithm and prove its associated rersults in Proposition 1. It have been shown by the recent work of Xu et al. (2021a) that the user/item embeddings optimized via the skip-gram negative sampling (SGNS) algorithm are the sufficient dimension reduction of a specific co-occurrence statistics. In particular, the SGNS objective function is given by:

$$
\mathbb{P}(O_{ui} = 1|u, i) = \sigma\big(\Phi(i)^\mathsf{T}\Phi(u)\big) = \frac{1}{1 + \exp\big(-\Phi(i)^\mathsf{T}\Phi(u)\big)},
$$

$$
\mathbb{P}(O_{ui} = 0|u, i) = \sigma\big(-\Phi(i)^\mathsf{T}\Phi(u)\big) = \frac{1}{1 + \exp\big(\Phi(i)^\mathsf{T}\Phi(u)\big)}, \tag{A.1}
$$

$$
\ell_{u,i} = -\log\mathbb{P}(O_{ui} = 1|u, i) + k \cdot \mathbb{E}_{k\sim\text{Neg}(\mathcal{I})}\big[\log\mathbb{P}(O_{ui} = 0|u, i)\big],
$$

where $\Phi(u)$, $\Phi(i)$ are the embeddings for the user $u$ and item $i$, $k$ is the number of negative samples, and $\text{Neg}(\mathcal{I})$ is the uniform negative-sampling distribution on the item set. The risk objective is thus give by: $\mathbb{E}_P R(\Phi) = \sum_{(u,i)\in\mathcal{D}}\ell_{u,i}$. The SGNS algorithm has been widely applied to train user and item embeddings for various recommendation tasks.

The *sufficient dimension reduction* (SDR), on the other hand, is inspired by the sufficient statistics of exponential family. Broadly speaking, when predicting $Y$ with $X$, if all the information in $X$ about $Y$ can be compressed into a dimension reduction $f(X)$, i.e. $Y \perp X \mid f(X)$ or $p(Y|X)$ and $p\big(Y|f(X)\big)$ are the same, then $f(X)$ is a sufficient dimension reduction (SDR). According

to Globerson and Tishby (2003), finding the SDR $f(X)$ is equivalent to solving a optimization problem:

$$\min_{q(y, f(x))} d_{KL}\big(p(Y, X) \,\|\, q(Y, f(X))\big), \tag{A.2}$$

where $d_{KL}$ is the Kullback-Leibler divergence.

*Proof.* Let $R_{ui} := \frac{N_{ui}}{N_u N_i}$ be the relatedness measure defined in Xu et al. (2021a), where $N_{ui}$ is the co-count of $(u, i)$, and $N_u$, $N_i$ are the individual counts of $u$ and $i$. The co-occurrence statistics is given by $C_{ui} := O_{ui} \,|\, R_{ui}$, which is exposure $O_{ui}$ condition on the relatedness measure. It is shown in the Claim 1 of Xu et al. (2021a) that:

$$\min_{\Phi(u), \Phi(i) \in \mathbb{R}^d} \mathbb{E}_P R(\Phi) = \min_{\Phi(u), \Phi(i) \in \mathbb{R}^d} \frac{1}{|\mathcal{D}|} \sum_{(u,i) \in \mathcal{D}} d_{KL}\big(p(O_{ui} \,|\, \Phi(u), \Phi(i)) \,\|\, p(O_{ui} \,|\, R_{ui})\big), \tag{A.3}$$

so $\Phi$ gives the SDR of $C_{ui}$ in $\mathbb{R}^d$, according to the definition of (A.2). Note that the terms in the RHS of (A.3) are all binary KL divergence, and it easy to verify that binary KL divergence satisfies: $d_{KL}(a\|c) + d_{KL}(c\|b) \geq d_{KL}(a\|b) + (c - a) \log \frac{c(1-b)}{(1-c)b}$. Recall that $C_{ui}(P)$ and $C_{ui}(Q)$ are the co-occurrence statistics under $P$ and $Q$, respectively. Therefore, if we let $\Phi^{(P)}$ and $\Phi^{(Q)}$ be the solution to (A.3) under $P$ and $Q$, by assuming $\mathcal{S}_P = \mathcal{S}_Q$, it holds that:

$$
\begin{aligned}
\inf_{\Phi} \big(\mathbb{E}_P R(\Phi) + \mathbb{E}_Q R(\Phi)\big) &\geq \inf_{\Phi} \mathbb{E}_P R(\Phi) + \inf_{\Phi} \mathbb{E}_P R(\Phi) \\
&\geq \frac{1}{|\mathcal{S}_P|} \sum_{(u,i) \in \mathcal{S}_P} d_{KL}\big(p(O_{ui} \,|\, \Phi^{(P)}(u), \Phi^{(P)}(i)) \,\|\, C_{ui}(P))\big) + \\
&\quad d_{KL}\big(p(O_{ui} \,|\, \Phi^{(Q)}(u), \Phi^{(Q)}(i)) \,\|\, C_{ui}(Q)\big) \\
&= \frac{1}{|\mathcal{S}_P|} \sum_{(u,i) \in \mathcal{S}_P} d_{KL}(C_{ui}(P) \,\|\, C_{ui}(Q)) + C,
\end{aligned}
$$

and $C = \inf_{\Phi} \big(\mathbb{E}_P R(\Phi) + \mathbb{E}_Q R(\Phi)\big)$ when $P = Q$ a.s. $\qquad\square$

## A.3 BACKGROUND ON THE WASSERSTEIN DISTANCE

Perhaps the two most common discrepancy measures on probabilities are $\phi$-divergence and integral probability metric (IPM). Obviously, $\phi$-divergence does not satisfy our need of characterizing domain transportation because it requires $\mathcal{S}_P \subseteq \mathcal{S}_Q$ (weak overlap) to hold in practice. IPM, on the other hand, provides the flexibility and is defined as

$$d_{\mathcal{G}}(P, Q) = \sup_{g \in \mathcal{G}} \big| \int g dP - \int g dQ \big|,$$

where $\mathcal{G}$ is a class of real-valued bounded measurable function. In this direction, we find the Wasserstein distance from the *optimal transportation* (OT) theory Rachev (1985) a good fit to our goal. Although the maximum mean discrepancy distance is also valid, its computation and related theoretical results rely on the specification of a reproducing kernel Hilbert space, whose usage is less common in the IR literature.

The origin of the Wasserstein distance is to give the optimal cost of transporting mass from one marginal distribution to another:

$$d_{w,c}(P, Q) = \arg\min_{\pi \in \Pi(P,Q)} \int_{\mathcal{D} \times \mathcal{D}} c(\mathbf{z}, \mathbf{z}') d\pi(P, Q),$$

where $\Pi(P, Q)$ is the set of all joint probability measure whose marginals are $P$ and $Q$, and $c(\cdot, \cdot)$ is the cost function. Clearly, $d_{w,c}(P, Q)$ relies on the geometry of $P$ and $Q$ if the cost function is well-chosen. For recommender system, $\mathbf{z}$ is often given by $(\mathbf{x}_u, \mathbf{x}_i)$ or $(\Phi(u), \Phi(i))$, so it is natural to use the $\|\cdot\|_2$ distance: $c(\mathbf{z}, \mathbf{z}') = \|\mathbf{z} - \mathbf{z}'\|_2$. We denote by $d_W$ for this particular cost function, which

is also referred to as the 1-Wasserstein distance. The classical Kantorovich-Rubinstein theorem Dudley (2018) connects the $d_W$ to IPM by proving the duality result of:

$$d_W = \sup_{g:\{\|g\|_L \leq 1\}} \int_{\mathcal{D}} g dP - \int_{\mathcal{D}} g dQ. \tag{A.4}$$

In the above formulation, the constant of 1 under $\sup$ is less critical since we can always rescale $g$. The class of Lipschitz-bounded function covers a wide selection of models including neural networks as discussed in Fazlyab et al. (2019).

### A.4    PROOF FOR THEOREM 1

We first state a proposition for the following proofs.

**Proposition A.1.** *Suppose $\ell_f$ is semi-continuous for all $f \in \mathcal{F}$, $P$ is any distribution supported on $\mathcal{Z}$, and $c(\mathbf{z}, \mathbf{z}') = \|\mathbf{z} - \mathbf{z}'\|_2$. It holds:*

$$\min_{Q:d_W(Q,P)\leq\rho} \mathbb{E}_Q R(f) = \sup_{\lambda \geq 0} \big\{ - \lambda\rho + \mathbb{E}_{\mathbf{z}\sim P}\big[\inf_{\tilde{\mathbf{z}}\in\mathcal{Z}}\{\ell_f(\tilde{\mathbf{z}}) + \lambda \cdot c(\mathbf{z}, \tilde{\mathbf{z}})\}\big]\big\}.$$

*Proof.* It is shown by Gao and Kleywegt (2016) that:

$$\max_{Q:d_W(Q,P)\leq\rho} \mathbb{E}_Q R(f) = \inf_{\lambda \geq 0} \big\{\lambda\rho + \mathbb{E}_{\mathbf{z}\sim P}\big[\sup_{\tilde{\mathbf{z}}\in\mathcal{Z}}\{\ell_f(\tilde{\mathbf{z}}) - \lambda \cdot c(\mathbf{z}, \tilde{\mathbf{z}})\}\big]\big\}.$$

Following the above result, we immediately have:

$$\min_{Q:d_W(Q,P)\leq\rho} \mathbb{E}_Q R(f) = - \max_{Q:d_W(Q,P)\leq\rho} -\mathbb{E}_Q R(f)$$

$$= - \inf_{\lambda \geq 0} \big\{\lambda\rho + \mathbb{E}_{\mathbf{z}\sim P}\big[\sup_{\tilde{\mathbf{z}}\in\mathcal{Z}}\{-\ell_f(\tilde{\mathbf{z}}) - \lambda \cdot c(\mathbf{z}, \tilde{\mathbf{z}})\}\big]\big\}$$

$$= \sup_{\lambda \geq 0} \big\{ - \lambda\rho - \mathbb{E}_{\mathbf{z}\sim P}\big[\sup_{\tilde{\mathbf{z}}\in\mathcal{Z}}\{-\ell_f(\tilde{\mathbf{z}}) - \lambda \cdot c(\mathbf{z}, \tilde{\mathbf{z}})\}\big]\big\}$$

$$= \sup_{\lambda \geq 0} \big\{ - \lambda\rho + \mathbb{E}_{\mathbf{z}\sim P}\big[\inf_{\tilde{\mathbf{z}}\in\mathcal{Z}}\{\ell_f(\tilde{\mathbf{z}}) + \lambda \cdot c(\mathbf{z}, \tilde{\mathbf{z}})\}\big]\big\}.$$

$\square$

Now we prove the result in Theorem 1.

*Proof.* We first define $\mathcal{H}_1 = \{w \in \mathcal{H} : d_W(P_w, P_f) \leq \rho\}$, and $\mathcal{H}_1 = \{w \in \mathcal{H} : d_W(P_w, P_f) \leq \rho + \frac{c_w}{n^{1/(d+1)}} + 8M\sqrt{\frac{\log\frac{4}{\delta}}{n}}\}$, where $c_w$ is the constant stated in Lemma A.5.

We then omit the $w \in \mathcal{H}$ in the following proofs for brevity. By Lemma A.5, for any $\delta > 0$, it holds with probability at least $1 - \delta$ that $\mathcal{H}_1 \subseteq \mathcal{H}_2$, which suggests that for any given $f \in \mathcal{F}$:

$$\min_{w:D(w,f)\leq\rho} \mathbb{E}_{P_w} R(f) \geq \min_{w:D_n(w,f)\leq\delta+B} \mathbb{E}_{P_w} R(f)$$

with probability at least $1 - \delta$, where $B = \frac{c_w}{n^{1/(d+1)}} + 8M\sqrt{\frac{\log\frac{4}{\delta}}{n}}$.

Now we define $\tilde{w} = \arg\min_{w:D_n(w,f)\leq\delta+B} \mathbb{E}_{P_w} R(f)$. Then using the result in Lemma A.1, for any given $f \in \mathcal{F}$, it holds with probability at least $1 - \frac{\delta}{4}$ that:

$$\min_{w:D_n(w,f)\leq\delta+B} \mathbb{E}_{P_w} R(f) = \mathbb{E}_{P_{\tilde{w}}} R(f)$$

$$\geq \mathbb{E}_{P_{n,\tilde{w}}} R(f) - \frac{7M\log\frac{1}{\delta}}{3n} - \sqrt{\frac{2R_{\tilde{w}}(f)^2 \log\frac{1}{\delta}}{n}}$$

$$\geq \min_{w:D_n(w,f)\leq\delta+B} \mathbb{E}_{P_{n,w}} R(f) - \frac{7M\log\frac{1}{\delta}}{3n} - \sqrt{\frac{2R_{\tilde{w}}(f)^2 \log\frac{1}{\delta}}{n}},$$

where the last line follows from the definition of $\tilde{w}$.

Using Proposition A.1, we have:

$$\min_{w:D_n(w,f)\leq\delta+B}\mathbb{E}_{P_{m,w}}R(f) = \sup_{\lambda\geq 0}\Big\{-\lambda(\rho+B)+\mathbb{E}_{\mathbf{z}\sim P_{n,w}}\big[\inf_{\tilde{\mathbf{z}}\in\mathcal{Z}}\{\ell_f(\tilde{\mathbf{z}})+\lambda\cdot c(\mathbf{z},\tilde{\mathbf{z}})\}\big]\Big\}$$

$$\geq \sup_{\lambda\geq 0}\Big\{-\lambda\rho+\mathbb{E}_{\mathbf{z}\sim P_{n,w}}\big[\inf_{\tilde{\mathbf{z}}\in\mathcal{Z}}\{\ell_f(\tilde{\mathbf{z}})+\lambda\cdot c(\mathbf{z},\tilde{\mathbf{z}})\}\big]\Big\} - L\cdot B$$

$$= \min_{w:D_n(w,f)\leq\delta}\mathbb{E}_{P_{n,w}}R(f) - L\cdot B,$$

where the second line uses the result in Lemma A.6 which bounds the $\lambda^*$ that achieves the supremum with the Lipscitize constant $L$. Combining the above results, it holds with probability at least $1-\frac{\delta}{2}$ that:

$$\min_{w:D(w,f)\leq\rho}\mathbb{E}_{P_w}R(f) \geq \min_{w:D_n(w,f)\leq\delta}\mathbb{E}_{P_{n,w}}R(f) - \frac{7M\log\frac{1}{\delta}}{3n} - \sqrt{\frac{2R_{\tilde{w}}(f)^2\log\frac{1}{\delta}}{n}} - L\cdot B.$$

The upper bound can be shown in the same manner, by using $\bar{w} = \arg\min_{w:D_n(w,f)\leq\delta-B}\mathbb{E}_{P_w}R(f)$ to replace $\tilde{w}$. Combining the upper and lower bounds, we obtain the result stated in Theorem 1.

$\square$

## A.5 PROOF FOR THEOREM 2

Suppose $\ell$ is some bounded loss function, then $M_{\mathcal{F}}$ stated in Theorem 2 must be finite for $\lambda\in[0,\frac{L}{\rho}]$, otherwise it can be shown from Proposition A.1 and Lemma A.6 that:

$$\min_{d_W(Q,P)\leq\rho}\mathbb{E}_Q R(f) = \sup_{\lambda\geq 0}\Big\{-\lambda\rho+\mathbb{E}_{\mathbf{z}\sim P}[\inf_{\tilde{\mathbf{z}}\in\mathcal{Z}}\ell_f(\tilde{\mathbf{z}})+\lambda\cdot c(\mathbf{z},\tilde{\mathbf{z}})]\Big\}$$

is unbounded for finite $\rho$, which contradicts that $\ell$ is bounded.

We first prove the generalization error bound.

*Proof.* We use the shorthand $\phi_{f,\lambda}(\mathbf{z}) := \mathbb{E}_{\mathbf{z}\sim P_f}[\inf_{\tilde{\mathbf{z}}}\ell_f(\tilde{\mathbf{z}})+\lambda c(\tilde{\mathbf{z}},\mathbf{z})]$. Since we do not make any parametric assumption on the weighting function $w$ here (as the consequence of using $w\in\mathcal{H}$ has been stated in Theorem 1), we have:

$$\min_{d_W(P_w,P_f)\leq\rho}\mathbb{E}_Q R(f) = \sup_{\lambda\geq 0}\Big\{-\lambda\rho+\mathbb{E}_{\mathbf{z}\sim P_f}\phi_{f,\lambda}(\mathbf{z})\Big\} \quad \text{By Proposition A.1}$$

$$= \sup_{\lambda\geq 0}\Big\{-\lambda\rho+\mathbb{E}_{\mathbf{z}\sim P_{n,f}}\phi_{f,\lambda}(\mathbf{z})+\mathbb{E}_{\mathbf{z}\sim P_f}\phi_{f,\lambda}(\mathbf{z})-\mathbb{E}_{\mathbf{z}\sim P_{n,f}}\phi_{f,\lambda}(\mathbf{z})\Big\}$$

$$\leq \sup_{\lambda\geq 0}\Big\{-\lambda\rho+\mathbb{E}_{\mathbf{z}\sim P_{n,f}}+\underbrace{\sup_{f\in\mathcal{F}}\big\{\mathbb{E}_{P_f}\phi_{f,\lambda}(\mathbf{z})-\mathbb{E}_{P_{n,f}}\phi_{f,\lambda}(\mathbf{z})\big\}}_{\psi_\lambda}\Big\}$$

Using the standard symmetrization argument, we can upper-bound $\mathbb{E}\psi_\lambda = \frac{1}{n}\sup_{f\in\mathcal{F}}\big\{\mathbb{E}_{P_f}\phi_{f,\lambda}(\mathbf{z})-\phi_{f,\lambda}(\mathbf{z}_i)\big\}$ with the empirical Rademarcher complexity:

$$\mathbb{E}\psi_\lambda \leq 2\mathbb{E}\sup_{f\in\mathcal{F}}\frac{1}{n}\sum_{i=1}^n\epsilon_i\phi_{f,\lambda}(\mathbf{z}_i),$$

where the expectation on LHS is taken with respect to the random variables of $\mathbf{z}_1,\ldots,\mathbf{z}_n$. Clearly, $\psi(\lambda)$ is bounded by $M_{\mathcal{F}}$ that we illustrated in the beginning, so from McDiarmid's inequality we have:

$$\mathbb{P}\big(\psi(\lambda)\geq\mathbb{E}\psi(\lambda)+\frac{Mt}{\sqrt{n}}\big)\leq\exp(-2t^2). \tag{A.5}$$

Also, note that $\sqrt{\frac{1}{n}}\epsilon_i\phi_{f,\lambda}(\mathbf{z}_i)$ is zero-mean and 1-subgaussian with respect to the $\|\cdot\|_\infty$ norm on $\mathcal{F}$, so for any $f_1, f_2 \in \mathcal{F}$, it holds that:

$$
\mathbb{E}\big[\exp\big(\frac{t}{n}\sum_{i=1}^n \epsilon_i(\phi_{f_1,\lambda}(\mathbf{z}_i) - \phi_{f_2,\lambda}(\mathbf{z}_i)))\big]
$$

$$
\leq \mathbb{E}\big[\exp\big(\frac{t}{\sqrt{n}}\epsilon \cdot \big(\inf_{\bar{\mathbf{z}}\in\mathcal{Z}}\{f_1(\bar{\mathbf{z}}) + \lambda c(\mathbf{z}_1, \bar{\mathbf{z}})\} + \sup_{\tilde{\mathbf{z}}\in\mathcal{Z}}\{-f_2(\tilde{\mathbf{z}}) - \lambda c(\mathbf{z}_1, \tilde{\mathbf{z}})\})\big)\big]^n \tag{A.6}
$$

$$
\leq \mathbb{E}\big[\exp\big(\frac{t}{\sqrt{n}}\epsilon\sup_{\mathbf{z}\in\mathcal{Z}}\{f_1(\mathbf{z}) - f_2(\mathbf{z})\}\big)\big]^n
$$

$$
\leq \exp\big(\frac{t^2\|f_1 - f_2\|_\infty}{2}\big).
$$

Using the entropy integral bound on the empirical Rademarcher complexity term, we have:

$$
\mathbb{E}\psi(\lambda) \leq \frac{c}{\sqrt{n}}\int_0^\infty \sqrt{\log\mathcal{N}(\epsilon, \mathcal{F}, \ell_\infty(P_n))}d\epsilon,
$$

since $\ell$ is Lipscitz (and we absorb the Lipscitz constant into $c$). Combining the above results, for any given $\lambda \geq 0$, it holds with probability at least $1 - \delta$ that:

$$
\mathbb{P}\Big(\exists f \in \mathcal{F} : \min_{d_W(P_w, P_f)\leq\rho}\mathbb{E}_Q R(f) \geq -\lambda\rho + \mathbb{E}_{\mathbf{z}\sim P_{n,f}}\big[\inf_{\tilde{\mathbf{z}}\in\mathcal{Z}}\ell_f(\tilde{\mathbf{z}}) + \lambda c(\mathbf{z}, \tilde{\mathbf{z}})\big] +
$$

$$
\frac{c\int_0^\infty\sqrt{\log\mathcal{N}(\epsilon, \mathcal{F}, \ell_\infty^n)}d\epsilon}{n} + \frac{M_\mathcal{F}t}{n}\Big) \leq 2\exp(-2t^2).
$$

Then we apply the same union-bound argument from Theorem 1 Koltchinskii et al. (2002) to extend the above result to any $\lambda > 0$. In particular, we pick the sequence of $\lambda_k = k$ and $t_k = t + \sqrt{\log k}$, $k = 1, 2, \ldots$. By simple calculations and the sum of infinite series arguments, we reach the conclusino that for any $f \in \mathcal{F}$, it holds with probability at least $1 - \delta$ that:

$$
\min_{d_W(P_w, P_f)\leq\rho}\mathbb{E}_Q R(f) \geq \min_{\lambda\geq 0}\Big\{-\lambda\rho + \mathbb{E}_{\mathbf{z}\sim P_{n,f}}\big[\inf_{\tilde{\mathbf{z}}\in\mathcal{Z}}\ell_f(\tilde{\mathbf{z}}) + \lambda c(\mathbf{z}, \tilde{\mathbf{z}})\big] +
$$

$$
\frac{c\int_0^\infty\sqrt{\log\mathcal{N}(\epsilon, \mathcal{F}, \ell_\infty^n)}d\epsilon}{n} + \frac{M_\mathcal{F}\sqrt{\log\frac{1}{2\delta}}}{n} + \sqrt{\frac{\log(\lambda+1)}{n}}\Big\}
$$

$\square$

Now we prove the excessive risk bound.

*Proof.* We start by defining: $\hat{f} = \arg\min_{f\in\mathcal{F}}\min_{D_n(w,f)\leq\rho}\mathbb{E}_{P_{n,w}}R(f)$, and $f^* = \arg\min_{f\in\mathcal{F}}\min_{D(w,f)\leq\rho}\mathbb{E}_{P_{n,w}}R(f)$. Using the definition of $\hat{f}$, we have:

$$
\min_{D(w,\hat{f})\leq\rho}\mathbb{E}_{P_w}R(\hat{f}) - \min_{D(w,f^*)\leq\rho}\mathbb{E}_{P_w}R(f^*)
$$

$$
\leq \min_{D(w,\hat{f})\leq\rho}\mathbb{E}_{P_w}R(\hat{f}) - \min_{D_n(w,\hat{f})\leq\rho}\mathbb{E}_{P_w}R(\hat{f}) + \min_{D_n(w,f^*)\leq\rho}\mathbb{E}_{P_w}R(f^*) - \min_{D(w,f^*)\leq\rho}\mathbb{E}_{P_w}R(f^*).
$$

Let $\hat{\lambda} = \arg\max_{\lambda\geq 0}\big\{-\lambda\rho + \mathbb{E}_{\mathbf{z}\sim P_{\hat{f}}}[\inf_{\tilde{\mathbf{z}}\in\mathcal{Z}}\ell_{\hat{f}}(\tilde{\mathbf{z}}) + \lambda\cdot c(\mathbf{z}, \tilde{\mathbf{z}})]\big\}$, and from Lemma A.6 we know that $\hat{\lambda} \leq L$. It holds that:

$$
\min_{D(w,\hat{f})\leq\rho}\mathbb{E}_{P_w}R(\hat{f}) - \min_{D_n(w,\hat{f})\leq\rho}\mathbb{E}_{P_w}R(\hat{f})
$$

$$
= \Big(-\hat{\lambda}\rho + \mathbb{E}_{\mathbf{z}\sim P_f}[\inf_{\bar{\mathbf{z}}\in\mathcal{Z}}\ell_{\hat{f}}(\tilde{\mathbf{z}}) + \hat{\lambda}\cdot c(\mathbf{z}, \tilde{\mathbf{z}})]\Big) - \max_{\lambda\geq 0}\Big\{-\lambda\rho + \mathbb{E}_{\mathbf{z}\sim P_f}[\inf_{\bar{\mathbf{z}}\in\mathcal{Z}}\ell_{\hat{f}}(\tilde{\mathbf{z}}) + \lambda\cdot c(\mathbf{z}, \tilde{\mathbf{z}})]\Big\}
$$

$$
\leq \int_\mathcal{Z}\big(\inf_{\tilde{\mathbf{z}}}\ell_{\hat{f}}(\tilde{\mathbf{z}}) + \hat{\lambda}c(\mathbf{z}, \tilde{\mathbf{z}})\big)d\big(P_{\hat{f}} - P_{n,\hat{f}}\big)
$$

$$
\leq \sup_{\lambda\in[0,L], \psi\in\ell\circ\mathcal{F}}\int_\mathcal{Z}\big(\inf_{\tilde{\mathbf{z}}}\ell_{\hat{f}}(\tilde{\mathbf{z}}) + \lambda c(\mathbf{z}, \tilde{\mathbf{z}})\big)d\big(P_{\hat{f}} - P_{n,\hat{f}}\big).
$$

Again applying the standard symmetrization argument and McDiarmid's inequality to the last line (similar to what we have shown in proving the generalization bound), the following bound holds with probability at least $1 - \delta/2$:

$$\min_{D(w,\hat{f}) \leq \rho} \mathbb{E}_{P_w} R(\hat{f}) - \min_{D_n(w,\hat{f}) \leq \rho} \mathbb{E}_{P_w} R(\hat{f}) \leq 2Rad_n\big([0,L] \otimes \ell \circ \mathcal{F}\big) + M_{\mathcal{F}} \sqrt{\frac{2\log\frac{2}{\delta}}{n}}.$$

Similarly, we let $\lambda^* = \arg\max_{\lambda \geq 0} \big\{ -\lambda\rho + \mathbb{E}_{\mathbf{z} \sim P_{f^*}}[\inf_{\tilde{\mathbf{z}} \in \mathcal{Z}} \ell_{f^*}(\tilde{\mathbf{z}}) + \lambda \cdot c(\mathbf{z}, \tilde{\mathbf{z}})] \big\}$, and using the same arguments, we also have the following bound that holds with probability at least $1 - \delta/2$:

$$\min_{D(w,f^*) \leq \rho} \mathbb{E}_{P_w} R(f^*) - \min_{D_n(w,f^*) \leq \rho} \mathbb{E}_{P_w} R(f^*) \leq 2Rad_n\big([0,L] \otimes \ell \circ \mathcal{F}\big) + M_{\mathcal{F}} \sqrt{\frac{2\log\frac{2}{\delta}}{n}}.$$

Combining the above two bounds, we obtain the desired result. $\qquad\square$

## A.6 PROOF FOR CLAIM 1

*Proof.* Given $f \in \mathcal{F}$, the mapping of $P_w$ to $d_W(P_w, P_f)$ is convex in the space of probability measures. Since $d_W(P_w, P_f) = 0$ can be achieved by taking $P_w = P_f$ a.e, using the generalized Lagrange duality result in Lemma A.3, it holds that:

$$\inf_{d_W(P_w,P_f) \leq \rho} \int_{\mathcal{Z}} \ell_f(\mathbf{z}) dP_w = \sup_{\lambda \geq 0} \inf_{d_W(P_w,P_f) \leq \rho} \big\{ -\lambda\rho + \int_{\mathcal{Z}} \ell_f(\mathbf{z}) dP_w + \lambda \cdot d_W(P_w, P_f) \big\}.$$

Notice that for any $\pi \in \Pi(P_w, P_f)$, where $\Pi(P_w, P_f)$ is the set of all joint distributions whose marginals are given by $P_w$ and $P_f$, we have: $\int_{\mathcal{Z}} \ell_f(\mathbf{z}) dP_w = \int_{\mathcal{Z}} \int_{\mathcal{Z}} \ell_f(\mathbf{z}_1) d\pi(\mathbf{z}_1, \mathbf{z}_2)$. Therefore, we have:

$$\int_{\mathcal{Z}} \ell_f(\mathbf{z}) dP_w + \inf_{\pi \in \Pi(P_w,P_f)} \int_{\mathcal{Z} \times \mathcal{Z}} c(\mathbf{z}_1, \mathbf{z}_2) d\pi(\mathbf{z}_1, \mathbf{z}_2) = \inf_{\pi \in \Pi(P_w,P_f)} \big\{ \int_{\mathcal{Z} \times \mathcal{Z}} \ell_f(\mathbf{z}_1) + \lambda \cdot c(\mathbf{z}_1, \mathbf{z}_2) \big\},$$

using the definition of the Wasserstein distance introduced in Appendix A.3. As a consequence, it holds that:

$$\inf_{d_W(P_w,P_f) \leq \rho} \int_{\mathcal{Z}} \ell_f(\mathbf{z}) dP_w = \sup_{\lambda \geq 0} \inf_{w,\pi \in \Pi(P_w,P_f)} \big\{ -\lambda\rho + \int_{\mathcal{Z} \times \mathcal{Z}} \ell_f(\mathbf{z}_1) + \lambda \cdot c(\mathbf{z}_1, \mathbf{z}_2) \big\}.$$

We first point out that:

$$\inf_{w,\pi \in \Pi(P_w,P_f)} \big\{ \int_{\mathcal{Z} \times \mathcal{Z}} \ell_f(\mathbf{z}_1) + \lambda \cdot c(\mathbf{z}_1, \mathbf{z}_2) \big\} \geq \int_{\mathcal{Z}} \inf_{\mathbf{z}_1} \big\{ \ell_f(\mathbf{z}_1) + \lambda \cdot c(\mathbf{z}_1, \mathbf{z}_2) \big\} dP_f(\mathbf{z}_2), \quad \text{(A.7)}$$

and in the following proof we show that the other direction also holds, so the above inequality becomes equality. Also, if we consider $\mathcal{M}$ as the space of all the regular conditional probabilities from $\mathbf{z}_2$ to $\mathbf{z}_1$, it is evident that:

$$\inf_{w,\pi \in \Pi(P_w,P_f)} \big\{ \int_{\mathcal{Z}} \ell_f(\mathbf{z}_1) + \lambda \cdot c(\mathbf{z}_1, \mathbf{z}_2) dP_f(\mathbf{z}_2) \big\}$$
$$\leq \inf_{M \in \mathcal{M}} \big\{ \int_{\mathcal{Z}} \ell_f(\mathbf{z}_1) + \lambda \cdot c(\mathbf{z}_1, \mathbf{z}_2) dM(\mathbf{z}_1|\mathbf{z}_2) dP_f(\mathbf{z}_2) \big\}. \quad \text{(A.8)}$$

Furthermore, using Lemma A.8, if we consider $\mathcal{X}$ as the space of all measurable mappings from $\mathcal{Z}$ to $\mathcal{Z}$, then it holds:

$$\inf_{x \in \mathcal{X}} \big\{ \int_{\mathcal{Z}} \ell_f(x(\mathbf{z})2)) + \lambda c(x(\mathbf{z}_2), \mathbf{z}_2) dP_f(\mathbf{z}_2) \big\} = \int_{\mathcal{Z}} \inf_{\mathbf{z}_1} \big\{ \ell_f(\mathbf{z}_1) + \lambda \cdot c(\mathbf{z}_1, \mathbf{z}_2) \big\} dP_f(\mathbf{z}_2).$$

Consequently, if we let $\tilde{x} \in \mathcal{X}$ be the mapping that achieves $\epsilon$-close to the above infimum (which requires certain continuity condition for $\ell_f$) and $M(\cdot|\mathbf{z}_2)$ be defined by $\tilde{x}(\mathbf{z}_2)$:

$$\int_{\mathcal{Z}} \ell_f(\mathbf{z}_1) + \lambda c(\mathbf{z}_1, \mathbf{z}_2) dM(\mathbf{z}_1|\mathbf{z}_2) dP_f(\mathbf{z}_2) = \int_{\mathcal{Z}} \ell_f(\tilde{x}(\mathbf{z}_2)) + \lambda c(\tilde{\mathbf{z}}_2, \mathbf{z}_2) dP_f(\mathbf{z}_2)$$
$$\leq \int_{\mathcal{Z}} \inf_{\mathbf{z}_1} \big\{ \ell_f(\mathbf{z}_1) + \lambda \cdot c(\mathbf{z}_1, \mathbf{z}_2) \big\} dP_f(\mathbf{z}_2) + \epsilon$$
$$\leq \inf_{w,\pi \in \Pi(P_w,P_f)} \big\{ \int_{\mathcal{Z}} \ell_f(\mathbf{z}_1) + \lambda \cdot c(\mathbf{z}_1, \mathbf{z}_2) dP_f(\mathbf{z}_2) \big\} + \epsilon$$
$$\text{(A.9)}$$

Since $\epsilon > 0$ is arbitrary, combining (A.7), (A.8) and (A.9), we have:

$$\inf_{w,\pi \in \Pi(P_w, P_f)} \left\{ \int_{\mathcal{Z}} \ell_f(\mathbf{z}_1) + \lambda \cdot c(\mathbf{z}_1, \mathbf{z}_2) dP_f(\mathbf{z}_2) \right\} = \int_{\mathcal{Z}} \inf_{\mathbf{z}_1} \left\{ \ell_f(\mathbf{z}_1) + \lambda \cdot c(\mathbf{z}_1, \mathbf{z}_2) \right\} dP_f(\mathbf{z}_2),$$

and using the result in Proposition A.1, we obtain the final that for any $f \in \mathcal{F}$:

$$\min_{d_W(P_w, P_f) \leq \rho} \mathbb{E}_{P_w} R(f) = \max_{\lambda \geq 0} \left\{ -\lambda\rho + \min_{P_w} \{ \mathbb{E}_{P_w} R(f) + \lambda \cdot d_W(P_w, P_f) \} \right\}.$$

$\square$

### A.7 PROOF FOR THEOREM 3

*Proof.* We first let $f^* = \arg\min_{f \in \mathcal{F}} \mathbb{E}_{P_f} R(f)$, and define $w^* = f^*$ (suppose $\mathcal{F} \subset \mathcal{H}$) such that $P_{n,w^*}(u,i) = P_{n,f^*}(u,i)$ and $P_{w^*}(u,i) = P_{f^*}(u,i)$. Since $\mathcal{F}$ is the class of valid exploration policies in the bandit setting, we have $f(u,i) \in (0,1)$ for all $f \in \mathcal{F}$. Note that by the definition of $f^*$ and $w^*$:

$$\min_{f \in \mathcal{F}, h \in \mathcal{H}} \mathbb{E}_{P_{n,w}} R(f) + \lambda \cdot d_W(P_{n,w}, P_{n,f}) \leq \mathbb{E}_{P_{n,w^*}} R(f^*) + \lambda \cdot d_W(P_{n,w^*}, P_{n,f^*}).$$

Combining the results in Lemma A.1 and Lemma A.5, it holds with probability at least $1 - \delta/2$ for any $f \in \mathcal{F}$ and $w \in \mathcal{H}$ that:

$$\begin{cases} \mathbb{E}_{P_{n,w^*}} R(f^*) - \mathbb{E}_{P_{w^*}} R(f^*) \leq \frac{7 \log \frac{1}{4\delta}}{3n} + \sqrt{\frac{2 R_{w^*}(f^*)^2 \log \frac{1}{4\delta}}{n}}, \\ d_W(P_{n,w^*}, P_{n,f^*}) \leq \frac{c_w}{n^{1/(d+1)}} + 8 M_G \sqrt{\frac{\log \frac{1}{\delta}}{n}} \end{cases}$$

Hence, the following inequality holds with probability at least $1 - \delta$:

$$\min_{f \in \mathcal{F}, h \in \mathcal{H}} \mathbb{E}_{P_{n,w}} R(f) + \lambda \cdot d_W(P_{n,w}, P_{n,f})$$

$$\leq \min_{f \in \mathcal{F}} \mathbb{E}_{P_f} R(f) + \frac{7 \log \frac{1}{2\delta}}{3n} + \sqrt{\frac{2 R_{w^*}(f^*)^2 \log \frac{1}{2\delta}}{n}} + \frac{c_w}{n^{1/(d+1)}} + 8 M_G \sqrt{\frac{\log \frac{2}{\delta}}{n}}. \quad (A.10)$$

On the other hand, with $\lambda \gtrsim L \geq 1$ where $L$ is the Lipschitz constant for $\ell \circ \mathcal{F}$, for any given $f \in \mathcal{F}$ and $w \in \mathcal{H}$, it holds that:

$$\mathbb{E}_{P_f} R(f) = \mathbb{E}_{P_w} R(f) + \int_{\mathcal{Z}} \ell_f(\mathbf{z}) \left( -\frac{P_w(\mathbf{z})}{\pi(\mathbf{z})} + \frac{P_f(\mathbf{z})}{\pi(\mathbf{z})} \right) d\mathbf{z}$$

$$\lesssim \mathbb{E}_{P_w} R(f) + \sup_{\psi \in \ell \circ \mathcal{F}} \mathbb{E}_{P_w} \psi(\mathbf{z}) - \mathbb{E}_{P_f} \psi(\mathbf{z})$$

$$\leq \mathbb{E}_{P_w} R(f) + L \cdot d_W(P_w, P_f),$$

where in the first line we use the background of learning from bandit feedback data where $\pi$ is the logging policy (e.g. $\pi(\mathbf{z}) = \pi(u,i)$ for $\mathbf{z} = (\mathbf{x}_u, \mathbf{x}_i, y_{ui})$); and the last line we use the definition of Wasserstein distance under the Kantorovich-Rubinstein theorem (Appendix A.3). Again, by combining Lemma A.1 and Lemma A.5, we can further bound the last line with probability at least $1 - \delta/2$ with:

$$\mathbb{E}_{P_{n,w}} R(f) + L \cdot d_W(P_{n,w}, P_{n,f}) + \lambda \underbrace{\left( \frac{7 \log \frac{1}{4\delta}}{3n} + \sqrt{\frac{2 R_w(f)^2 \log \frac{1}{4\delta}}{n}} + \frac{c_w}{n^{1/(d+1)}} + 8 M_G \sqrt{\frac{\log \frac{1}{\delta}}{n}} \right)}_{\text{slack terms}}.$$

Now we let $\tilde{f}, \tilde{w} = \arg\min_{f \in \mathcal{F}, w \in \mathcal{H}} \mathbb{E}_{P_{n,w}} R(f) + \lambda \cdot d_W(P_{n,w}, P_{n,f})$ for $\lambda \gtrsim L \geq 1$. Then it holds that:

$$\min_{f \in \mathcal{F}} \mathbb{E}_{P_f} R(f) \leq \mathbb{E}_{P_{\tilde{f}}} R(\tilde{f}) \quad (A.11)$$

$$\leq \mathbb{E}_{P_{n,\tilde{w}}} R(\tilde{f}) + \lambda \cdot d_W(P_{n,\tilde{w}}, P_{n,\tilde{f}}) + \text{slack terms},$$

where the slack terms are stated above. Combining (A.10) and (A.11), we obtain the desired result.

$\square$

A.8  ANALYSIS ON THE GDA ALGORITHM.

By using bounded Lipschitz functions for $w_{\boldsymbol{\theta}_1}$, $f_{\boldsymbol{\theta}_2}$, we will show below that the gradient of $\mathcal{L}_\lambda$ is $L$−Lipschitz. It is indicated by Daskalakis and Panageas (2018) that GDA with $\eta \leq 1/L$ avoids unstable critical points for almost all initialization, which means that $\left(\boldsymbol{\theta}_1^{(t}, \boldsymbol{\theta}_2^{(t)}, \boldsymbol{\theta}_3^{(t)}\right)$ is unlikely to cycle as $t$ grows. Furthermore, any stationary point yielded by GDA with large $\gamma$ is basically a *local minimax point* except degenerated case (Jin et al., 2020), and thus the convergence guarantee of Algorithm 1. Here, a solution $(\boldsymbol{\theta}_1^*, \boldsymbol{\theta}_2^*, \boldsymbol{\theta}_3^*)$ is local minimax if there exits $\delta_0 > 0$ and a function $h$ satisfying $\lim_{\delta \to 0} h(\delta) \to 0$ such that $\forall \delta \in (0, \delta_0]$, and any $(\boldsymbol{\theta}_1, \boldsymbol{\theta}_2, \boldsymbol{\theta}_3)$ satisfying $\left\|[\boldsymbol{\theta}_1, \boldsymbol{\theta}_2] - [\boldsymbol{\theta}_1^*, \boldsymbol{\theta}_2^*]\right\|_2 \leq \delta$ and $\|\boldsymbol{\theta}_3 - \boldsymbol{\theta}_3^*\|_2 \leq h(\delta)$, it holds that:

$$\mathcal{L}_\lambda\big([\boldsymbol{\theta}_1^*, \boldsymbol{\theta}_2^*], \boldsymbol{\theta}_3\big) \leq \mathcal{L}_\lambda\big([\boldsymbol{\theta}_1^*, \boldsymbol{\theta}_2^*], \boldsymbol{\theta}_3^*\big) \leq \max_{\boldsymbol{\theta}_3': \|\boldsymbol{\theta}_3' - \boldsymbol{\theta}_3^*\|_2 \leq h(\delta)} \mathcal{L}_\lambda\big([\boldsymbol{\theta}_1, \boldsymbol{\theta}_2], \boldsymbol{\theta}_3'\big).$$

In other words, by using the two-time-scale GDA algorithm with a small $\eta$ and a large $\gamma$, our resulting solution has a similar role as a local minimum/maximum in a minimization/maximization problem — the global Nash equilibrium, if any, must be one of the solutions yielded by the GDA in Algorithm 1.

*Proof.* Here, we show that if $w_{\boldsymbol{\theta}_1}$, $f_{\boldsymbol{\theta}_2}$ are bounded Lipschitz functions and $g_{\boldsymbol{\theta}_3}$ is a Lipschitz function, the gradient of the minimax objective is also Lipschitz.

Recall the objective function in (6), i.e.,

$$\mathcal{L}_\lambda([\boldsymbol{\theta}_1, \boldsymbol{\theta}_2], \boldsymbol{\theta}_3) := \sum_{(u,i) \in \mathcal{D}} w_{ui}(\boldsymbol{\theta}_2) \ell_{ui}(f_{\boldsymbol{\theta}_1})$$

$$+ \lambda \left( \sum_{(u,i) \in \mathcal{D}} w_{ui}(\boldsymbol{\theta}_2) \cdot g_{ui}(\boldsymbol{\theta}_3) - \sum_{(u,i) \in \mathcal{D}} \text{reco}(u, i; f_{\boldsymbol{\theta}_i}) \cdot g_{ui}(\boldsymbol{\theta}_3) \right).$$

It is straightforward to show that:

$$\frac{\partial \mathcal{L}_\lambda}{\partial \boldsymbol{\theta}_1} = \sum_{(u,i) \in \mathcal{D}} w_{ui}(\boldsymbol{\theta}_2) \frac{\partial \ell_{ui}(f_{\boldsymbol{\theta}_1})}{\partial \boldsymbol{\theta}_1} - \lambda \sum_{(u,i) \in \mathcal{D}} g_{ui}(\boldsymbol{\theta}_3) \cdot \frac{\partial \text{reco}(u, i; f_{\boldsymbol{\theta}_1})}{\partial \boldsymbol{\theta}_1}$$

$$\frac{\partial \mathcal{L}_\lambda}{\partial \boldsymbol{\theta}_2} = \sum_{(u,i) \in \mathcal{D}} \ell_{ui}(f_{\boldsymbol{\theta}_1}) \frac{\partial w_{ui}(\boldsymbol{\theta}_2)}{\partial \boldsymbol{\theta}_2} + \lambda \sum_{(u,i) \in \mathcal{D}} g_{ui}(\boldsymbol{\theta}_3) \cdot \frac{\partial w_{ui}(\boldsymbol{\theta}_2)}{\partial \boldsymbol{\theta}_2}$$

$$\frac{\partial \mathcal{L}_\lambda}{\partial \boldsymbol{\theta}_3} = \lambda \left( \sum_{(u,i) \in \mathcal{D}} w_{ui}(\boldsymbol{\theta}_2) \cdot \frac{\partial g_{ui}(\boldsymbol{\theta}_3)}{\partial \boldsymbol{\theta}_3} - \sum_{(u,i) \in \mathcal{D}} \text{reco}(u, i; f_{\boldsymbol{\theta}_1}) \cdot \frac{\partial g_{ui}(\boldsymbol{\theta}_3)}{\partial \boldsymbol{\theta}_3} \right).$$

We impose a few assumptions on $f_{\boldsymbol{\theta}_1}$, $w_{ui}(\boldsymbol{\theta}_2)$ and $g_{ui}(\boldsymbol{\theta}_3)$. First, we assume they are bounded and their gradients are Lipschitz functions, which are standard assumptions for neural or factorization models. Then, with a careful selection of the loss function $\ell$ (e.g., logistic loss, exp-loss, probit loss) and the reco function (e.g., top-K), we can ensure $\ell_{ui}(f_{\boldsymbol{\theta}_1})$ and $\text{reco}(u, i; f_{\boldsymbol{\theta}_1})$ to have Lipschitz gradients. Finally, by employing clipping tricks on the gradients to ensure they will not explode, which is also a common practice, we have universally bounded and Lipschitz functions that include:

$f_{\boldsymbol{\theta}_1}$, $w_{ui}(\boldsymbol{\theta}_2)$, $g_{ui}(\boldsymbol{\theta}_3)$, $\frac{\partial \ell_{ui}(f_{\boldsymbol{\theta}_1})}{\partial \boldsymbol{\theta}_1}$, $\frac{\text{reco}(u,i;f_{\boldsymbol{\theta}_1})}{\partial \boldsymbol{\theta}_1}$, $\frac{\partial w_{ui}(\boldsymbol{\theta}_2)}{\partial \boldsymbol{\theta}_2}$, $\frac{g_{ui}(\boldsymbol{\theta}_3)}{\partial \boldsymbol{\theta}_3}$.

Therefore, the products and summations of these function is a $L$-Lipschitz function for some positive constant $L$. □

# B  LEMMAS AND PROOFS

In this part of the appendix, we state the important lemmas that we used in our proofs.

**Lemma A.1** (Empirical Bernstein bound). *Suppose that $\ell_f : \mathcal{Z} \to [0, 1]$ without loss of generality. With $w(u, i) = \frac{Q(u,i)}{P(u,i)}$ and $\mathbf{z}_1, \ldots, \mathbf{z}_n$ sampled i.i.d from $P$, for any $\delta > 0$, it holds with probability*

*at least $1 - \delta$ that:*

$$\mathbb{E}_Q R(f) \leq \mathbb{E}_{P_{n,w}} + \frac{7M \log \frac{1}{\delta}}{3n} + \sqrt{\frac{2R_w(f)^2 \log \frac{1}{\delta}}{n}},$$

*where $R_w(f)^2 = \frac{1}{n} \sum w_i^2 \ell_f(\mathbf{z}_i)^2$.*

The result is modified from Corollary 5 of Maurer and Pontil (2009).

**Lemma A.2.** *Let $P$ and $Q$ be the training and target distribution supported on $\mathcal{S}_P, \mathcal{S}_Q \subseteq \mathcal{D}$, and $w(u, i) = \frac{Q(u,i)}{P(u,i)}$, for $(u, i) \in \mathcal{S}_P \cap \mathcal{S}_Q$. $\mathcal{D}_n$ consists of training instances sampled i.i.d from $P$. Given a single hypothesis $f \in \mathcal{F}$, suppose $w_{ui} \in (0, M)$, for any $\delta > 0$, it holds with probability at least $1 - \delta$ that:*

$$\mathbb{E}_Q R(f) \leq \mathbb{E}_{P_{n,w}} R(f) + \frac{2M \log \frac{1}{\delta}}{3n} + \sqrt{\frac{2d(P\|Q) \log \frac{1}{\delta}}{n}},$$

*where $d(P\|Q) = \int_{\mathcal{S}_Q} (dP/dQ) dP$.*

The above result for importance weighting of a single hypothesis is stated in the Theorem 1 of Cortes et al. (2010).

**Lemma A.3.** *Let $f$ be a real-valued convex functional defined on a convex subset $\Omega$ of a vector space $\mathcal{X}$, and let $G$ be a convex mapping of $\mathcal{X}$ into a normed space $\mathcal{Z}$. Suppose there exists $\mathbf{x}_1$ such that $G(\mathbf{x}_1) < \rho$ and that $\inf\{f(\mathbf{x}) : G(\mathbf{x}) \leq \rho, \mathbf{x} \in \Omega\}$ is finite, then it holds that:*

$$\inf_{G(\mathbf{x}) \leq \rho, \mathbf{x} \in \Omega} = \max_{\mathbf{z}^* \geq \rho, \mathbf{z} \in \mathcal{Z}} \inf_{\mathbf{x} \in \Omega} \{f(\mathbf{x}) + \langle G(\mathbf{x}), \mathbf{z}^* \rangle\}.$$

The above Lagrange duality result is stated in Theorem 8.1 of Luenberger (1997).

**Lemma A.4** (Lower bound based on maximum variance). *Suppose $f \in \mathcal{F}$ takes values in $[0, 1]$ without loss of generality, and the variance of $f$ is denoted by $\sigma^2(f)$. Let $\sigma(\mathcal{F}) = \sup_{f \in \mathcal{F}} \sigma(f)$, and assume $\frac{1}{n} \leq \sigma^2(g) < +\infty$. It holds that:*

$$\mathbb{P}\left(\sup_{f \in \mathcal{F}} \frac{\mathbb{E}_P f - \mathbb{E}_{P_n} f}{\sigma(\mathcal{F})} \geq \frac{1}{2\sqrt{m}}\right) \geq c > 0.$$

The result is stated in Theorem 9 of Cortes et al. (2010)

**Lemma A.5** (Finite sample bound for 1-Wasserstein distance). *Suppose $P$ and $Q$ are supported on (perhaps non-overlapping subsets of) $\mathcal{Z}$ where $\mathcal{Z} \in \mathbb{R}^d$ and is bounded. Let $M_G := \sup_{g:\|g\|_L \leq 1} \sup_{\mathbf{z} \in \mathcal{Z}} g(\mathbf{z})$. For all $\delta > 0$, it holds with probability at least $1 - \delta$ that:*

$$\left| d_W(P, Q) - d_W(P_n, Q_n) \right| \leq \frac{c_w}{n^{1/(d+1)}} + 8M_G \sqrt{\frac{\log \frac{4}{\delta}}{n}}.$$

*Proof.* The first part of the proof follows from the result on integral probability metric stated in Theorem 11 of Sriperumbudur et al. (2009), that for $\mathcal{G} = \{g : \|g\|_L \leq 1\}$, we have:

$$\left| \sup_{g \in \mathcal{G}} \left[ \int_{\mathcal{Z}} g dP - \int_{\mathcal{Z}} g dQ \right] - \sup_{g \in \mathcal{G}} \left[ \int_{\mathcal{Z}} g dP_n - \int_{\mathcal{Z}} g dQ_n \right] \right| \leq 6\sqrt{\frac{2M_G^2 \log \frac{4}{\delta}}{n}} + 4\mathcal{R}_n(\mathcal{G}),$$

with probability at least $1 - \delta$, and $\mathcal{R}_n(\mathcal{G})$ is the empirical Rademarcher complexity of $\mathcal{G}$. Using the entropy integral bound on the empirical Rademarcher complexity, we have:

$$\mathcal{R}_n(\mathcal{G}) \leq 2\epsilon + \frac{4\sqrt{2}}{\sqrt{n}} \int_{\epsilon/4}^{\infty} \sqrt{\log \mathcal{N}(u, \mathcal{G}, \ell_2(P_n))} du.$$

Note that $\mathcal{Z} \in \mathbb{R}^d$ implies: $\mathcal{N}(u, \mathcal{Z}, \|\cdot\|_\infty) = \mathcal{O}(u^{-d})$. Using the fact that $\log(x + 1) \leq x + 1$, we have: $\log \mathcal{N}(u, \mathcal{G}, \ell_2(P_n)) \leq \log \mathcal{N}(u, \mathcal{G}, \|\cdot\|_\infty) \leq c_1 u^{-(d+1)} + c_2 u^{-d}$. Since $\mathcal{Z}$ is bounded, the entropy integral bound becomes:

$$\mathcal{R}_n(\mathcal{G}) \leq \inf_{\epsilon > 0} 2\epsilon + \frac{4\sqrt{2}}{\sqrt{n}} \int_{\epsilon/4}^{4\mathrm{diam}(\mathcal{Z})} \sqrt{c_1 u^{-(d+1)} + c_2 u^{-d}} du,$$

which directly leads to the desired bound where $c_w$ is the corresponding constant. $\qquad\square$

**Lemma A.6.** *Let* $c(\mathbf{z}, \tilde{\mathbf{z}}) := \|\mathbf{z} - \tilde{\mathbf{z}}\|_2$. *Define:*

$$f^* = \underset{f \in \mathcal{F}}{\arg\min} \ \underset{Q:d_W(Q,P) \leq \rho}{\min} \mathbb{E}_Q R(f), \ and \ \lambda^* = \underset{\lambda \geq 0}{\arg\max} \left\{ \lambda \rho + \mathbb{E}_{\mathbf{z} \sim P} \left[ \underset{\tilde{\mathbf{z}} \in \mathcal{Z}}{\inf} \ell_{f^*}(\tilde{\mathbf{z}}) + \lambda \cdot c(\mathbf{z}, \tilde{\mathbf{z}}) \right] \right\}.$$

*If all* $f \in \mathcal{F}$ *is L-Lipschitz, i.e.* $\sup_{\mathbf{z}, \mathbf{z}'} \frac{|f(\mathbf{z}) - f(\mathbf{z}')|}{c(\mathbf{z}, \mathbf{z}')} \leq L$, *then it holds:* $\lambda^* \leq L$.

*Proof.* Using the Lipschitz property, we have:

$$\lambda^* \rho \geq \lambda^* \rho + \underbrace{\mathbb{E}_{\mathbf{z} \sim P} \left[ \underset{\tilde{z} \in \mathcal{Z}}{\inf} \ell_{f^*}(\tilde{\mathbf{z}}) + \lambda^* \cdot c(\mathbf{z}, \tilde{\mathbf{z}}) - \ell_{f^*}(\mathbf{z}) \right]}_{\text{always} \leq 0, \text{because} = 0 \text{ when } \mathbf{z} = \tilde{\mathbf{z}}}$$

$$\geq -\lambda \rho + \mathbb{E}_{\mathbf{z} \sim P} \left[ \underset{\tilde{z} \in \mathcal{Z}}{\inf} \ell_{f^*}(\tilde{\mathbf{z}}) + \lambda^* \cdot c(\mathbf{z}, \tilde{\mathbf{z}}) - \ell_{f^*}(\mathbf{z}) \right] \quad (\text{ by the definition of } \lambda^*)$$

$$\geq -\lambda \rho + \mathbb{E}_{\mathbf{z} \sim P} \left[ -L \cdot c(\mathbf{z}, \tilde{\mathbf{z}}) + \lambda \cdot c(\mathbf{z}, \tilde{\mathbf{z}}) \right]$$

$$\geq -\lambda \rho + \underset{t \geq 0}{\inf} \{ -Lt + \lambda t \}$$

$$\geq -L\rho \quad (\text{ by choosing } \lambda = L),$$

which gives $\lambda^* \leq L$. $\qquad\qquad\qquad\qquad\qquad\qquad\qquad\qquad\qquad\qquad\qquad\qquad\qquad\square$

**Lemma A.7** (Change of measure inequality). *For any* $P$ *and* $Q$ *on* $\mathcal{X}$, *and for any measurable function* $f : \mathcal{X} \to \mathbb{R}$, *it holds that:*

$$\mathbb{E}_{x \sim Q} \leq d_{KL}(P\|Q) + \ln \left( \mathbb{E}_{x \sim P} e^{f(x)} \right).$$

This is a standard result that has been included in many textbooks for probability.

**Lemma A.8** (Interchange of integration and minimization). *Let* $\mathcal{X}$ *be a space of measurable functions from* $\mathcal{Z}$ *to* $\mathbb{R}^n$ *with respect to* $u$, *a* $\sigma$-*finite measure on the underlying space. Let* $f : \mathcal{Z} \times \mathbb{R}^n \to \mathbb{R}$ *be a normal integrand, then it holds:*

$$\underset{x \in \mathcal{X}}{\inf} \int_{\mathcal{Z}} f(z, x(z)) u(dz) = \int_{\mathcal{Z}} \underset{\mathbf{x} \in \mathbb{R}^n}{\inf} f(t, \mathbf{x}) u(dz)$$

The result is simplied from Theorem 14.60 of Rockafellar and Wets (2009).

## C COMPLETE EXPERIMENT SETTINGS, NUMERICAL RESULTS, AND REAL-WORLD DEPLOYMENT AND TESTING ANALYSIS

We provide the complete experiment settings (real-data and simulation experiments), including the data description and preprocessing, experiment setups, model configurations and training details. The complete numerical results and analysis are also presented in this part of the appendix. Finally, we discuss the real-world online experiments by deploying our solution to a e-commerce platform.

The implementation code for the real-data and simulation experiments are provided in the supplement material. All the reported results are averaged over ten independent runs.

### C.1 DATASET DESCRIPTION, PREPROCESSING AND EXPERIMENT SETUP

The detailed descriptions for the three real-world benchmark datasets are given as below.

- **Movielens-1M** [4]. The dataset provides the users' ratings for movies that they watched, which consists of around 1 million ratings collected from 60,40 users on 3,952 movies. The movie ratings are ranged from 1 to 5, where a higher rating indicates more positive feedback. Hence, the original Movielens-1M dataset possesses explicit feedback. The traditional way of converting the dataset to implicit feedback is to set the rating cut-off at 3, as suggested by He et al. (2017). We use the explicit feedback for semi-synthetic experiments, and use the converted explicit feedback for the real-data experiments.

---

[4]http://files.grouplens.org/datasets/movielens/ml-1m.zip

- **LastFM** [5]. The context of the LastFM dataset is music recommendation: each of the 1,892 listeners (recorded in the dataset) tags the artists they may find fond of overtime. Since the tag is a binary indicator, the LastFM is an implicit feedback dataset. There is a total of 186,479 tagging events, where 12,523 artists have been tagged.
- **GoodReads** [6]. The book recommendation dataset is obtained from the users' public shelves on *Goodread.com*. We use the user review data on the *history* and *biography* sections for their richness. There are in total 238,450 users, 302,346 unique books, and 2,066,193 ratings in these sections. The rating range is also from 1 to 5, and a higher rating indicates more positive feedback. We concert the dataset to implicit feedback in the same way as the Movielens-1M dataset for the real-data experiments.

**Data preprocessing**. As for preprocessing, we point out that the Movielens-1M dataset has been filtered by the publisher, where each user has rated at least 20 movies. For the LastFM and Goodread datasets, we first filter the infrequent items and users who have less than 20 total interactions as well.

### C.1.1 SIMULATION SETTING

We generate the implicit feedback (clicks) according to the click model of: $\mathbb{P}(Y_{ui} = 1) = \mathbb{P}(R_{ui} = 1) \cdot \mathbb{P}(O_{ui} = 1)$, where $R_{ui}$ and $O_{ui}$ indicates the relevance and exposure. We refer to $\mathbb{P}(R_{ui} = 1)$ as the relevance score. To carry over the inductive bias of the benchmark dataset to the generated data, we first learn $\mathbb{P}(R_{ui} = 1)$ and $\mathbb{P}(O_{ui} = 1)$ from the explicit feedback data of Movielens-1M and GoodReads. Our simulation approach does not apply to the LastFM data because it lacks the explicit feedback.

Since the ratings are explicit indicators for relevance, we directly conduct the matrix factorization collaborative filtering (MCF) using the ratings as the response. We then estimate $\mathbb{P}_{sim}(R_{ui} = 1)$ via $\sigma(\hat{R}_{ui} - u)^p$, where $\sigma(\cdot)$ is the sigmoid function, and $u$ and $p$ are the two hyperparameters that help us control the overall relevance distribution between the original and generated data. For both Movielens-1M and GoodReads, we find $u = 3$ and $p = 2$ gives a decent approximation. In this way, we inherit the inductive bias for the relevance to the simulated data.

As for the exposure, we first assume all the exposed movies/books are rated by the user, so we have $O_{ui} = 1$ if and only if $Y_{ui} > 0$. We then conduct another MCF using the converted $O_{ui}$ as response, and obtain the user and item embedding: $\Phi(u)$, $\Phi(i)$. We further design the exposure model by $\mathbb{P}_{sim}(O_{ui} = 1) = \sigma(\text{MLP}([\Phi(u), \Phi(i)]))$, where $\text{MLP}(\cdot)$ is a three-layer feed-forward neural network with ReLU activation and no bias terms, whose parameters are given by the random normal initialization.

In the sequel, the simulated click probability is given by: $\mathbb{P}_{sim}(Y_{ui} = 1) = \mathbb{P}_{sim}(R_{ui} = 1) \cdot \mathbb{P}_{sim}(O_{ui} = 1)$, from which we generated the click data by picking the cutoff probability that ensures the click v.s. non-click proportion of the generated data matches that of the original data.

### C.1.2 TRAIN-VALIDATION-TEST SPLIT

We adopt the widely-acknowledged approach for splitting the dataset to training, validation and testing as in He et al. (2017) and Rendle et al. (2020). In particular, we leverage the ordering information and use the last interaction of each user for testing, the second-to-last interaction for validation, and the previous interactions for training. All three datasets are provided with the ordering information, so we use the same splitting mechanism. For the semi-synthetic dataset, since the simulation does not account for the ordering information, we treat the ordering as random and conduct the splitting in the same fashion. We point out here that using sequential recommendation models is not reasonable in this case, so we do not experiment with the attention models (**Attn**) in the simulation experiments.

### C.1.3 EVALUATION

As we discussed in Section 6, the three metrics that we use for both the real-data and semi-synthetic experiments are **Rel@K**, **NDCG**@K and **Recall**@K. We also adopt the widely-acknowledged formulation of NDCG@K and Recall@K as from He et al. (2017) and Rendle et al. (2020): for

---

[5]http://files.grouplens.org/datasets/hetrec2011/hetrec2011-lastfm-2k.zip
[6]https://sites.google.com/eng.ucsd.edu/ucsdbookgraph/home

each positive instance (clicked data) of $(u', i', Y_{u'i'} = 1)$, we randomly sample 100 negative items $\{i_1, \ldots, i_{100}\}$ and compute NDCG@K and Recall@K according to where $i'$ is ranked among the total of 101 items. We point out that the recent work by Krichene and Rendle (2020) points out the potential bias issue of using the sampled metrics; however, it is still the mainstream approach that is used in the majority of existing literature. We apply them here so our results can be directly compared with the results in the relevant literature. To make up for the bias issue of using sampled metric, we further consider the Rel@K metric, which ranks all the items $\{i_1, \ldots, i_{|\mathcal{I}|}\}$ and see where $i'$ is placed for each $u'$. In this way, we eliminate the bias of using sampled metric. Essentially, Rel@K equals Recall@K with no sampling when each user interacts with only one item in the testing data (which happens in our experiments by the way we conduct the train-validation-test split).

The only difference in computing the three metrics for the real-data and semi-synthetic experiments is that we replace $Y_{u'i'}$ by the underlying relevance score of $\mathbb{P}_{\text{sim}}(R_{u'i'} = 1)$. In this way, we obtain a more accurate characterization of the actual testing performance. Finally, for the MovieLens and Goodreads datasets, we adopt the tradition that the previously-interacted items are not considered during the evaluation for each user. This is because in practice, the same movie or book should not be exposed to the user repeatedly.

## C.2 Model configuration and training

We provide the model configurations, training and computation details in this part of the appendix.

### C.2.1 Model Configurations

To achieve a fair comparison, we fix all the user and item embedding dimensions to $d = 32$ for the CF models and sequential recommendation model. For all the baseline models, we select the initial *learning rate* from $\{0.001, 0.005, 0.01, 0.05, 0.1\}$, and the $\ell_2$ regularization parameter from $\{0, 0.01, 0.05, 0.1, 0.2, 0.3\}$. The tuning parameters are selected separately to avoid excessive computations.

The other hyperparameters are identified according to the original papers as for the model-specific configurations, other than the embedding dimension, learning rate, and $\ell_2$ regularization. For instance, *NCF* considers the hidden dimensions of the feed-forward neural network as the hyperparameters, *IPW-CF* considers the IPW threshold, *Attn* considers the hidden dimension and number of attention blocks, *AC-MF* considers the adversarial strength of the IPW component. We use the same range as provided in their published implementations, which we discuss later in Appendix C.2.2. When implemented with our transportation-regularized risk minimization, we focus primarily on using the same "base model" for $f$, $w$ and $g$, e.g. with MCF, NCF or Attn. In fact, we can pick any combinations for $f$, $w$ and $g$, but we find the performances comparable to using the same "base model", so we randomly pick some of the configurations to report, instead of exhausting all the possibilities. Finally, the hyperparameters of the "base models" persist to our setting, in addition to the penalization parameter $\lambda$ (in eq.(4)), which we select from $\{0.005, 0.1, 0.3, 0.5\}$.

### C.2.2 Training details

For both the proposed approach and the baseline methods, we adopt the widely-applied negative-sampling-based training schema according to He et al. (2017) and Rendle et al. (2020), for both the real-data and simulation experiments. In particular, for each positive instance ($Y_{ui} = 1$), we randomly sample *three* pairs of user-item tuple and treat them as the negative instances. We point out that in many existing literature, the number of negative samples per positive instance is treated as a tuning parameter. Here, we treat it as a fixed quantity and focus on tuning the model-specific configurations.

For all the experiments, we use the *binary cross-entropy loss function* for classification, and apply *early stopping* during training when the Rel@K metric does not improve for more than five epochs. We record the model parameters after each epoch, and select the one that achieves the best validation performance for testing.

**Transportation-regularized risk minimization**. When training with the proposed approach, the discount of learning rate (in Algorithm 1) is critical for the two-time-scale GDA. In a sense it resembles the number of gradient updates under the same learning rate, and we focus on their relative

ratios. In practice, we allow the model of $w$ in the minimization step to have a learning rate different from that of $f$. We denote the ratio for $f$, $w$ and $g$ as $\text{step}_f$, $\text{step}_w$ and $\text{step}_g$. In practice, we select the ratio among $\{1:1:10, 1:5:10, 1:10:10, 1:10:5\}$ for our experiments, and the sensitivity analysis is provided in Appendix C.3.1.

**Baseline models**. The popularity (*Pop*) baseline is implemented with off-the-shelf code, the *MCF* and *NCF* models are implemented according to their original implementation published on Github[7], the IPW-debaised MF (*IPW-MF*) and user-exposure aware MF (*ExpoMF*) models are implemented using their published code[8] with necessary modifications. The adversarial-counterfactual MF (*AC-MF*) method is also implemented using the published code[9], which we apply directly to our datasets. The sequential recommendation model with attention mechanism is also implemented according to the original code[10]. We point out that most of the implementation provided by the authors include the set of hyper parameters they found optimal in their experiments. We re-select those hyper parameters in each of our settings.

As for the doubly robust joint learning (DR-MF) approach (Wang et al., 2019), the method was originally proposed for the bandit feedback setting or when the feedback data is explicit and the propensity scores can be effectively estimated. Since we focus on the implicit feedback setting, we use the clicks to estimate the propensities, and plug it to the DR approach. The causal embedding approach (CE-MF), on the other hand, assumes the access to a uniform exposure dataset. The authors propose a workaround to construct such data from non-uniform-exposure feedback data (Bonner and Vasile, 2018), however, they take advantage of the explicit feedback (rating). Since we only have implicit feedback, we random subsample the click data and treat them as if they are from the uniform exposure.

**Computation**. All the models are implemented with the auto-differentiation framework of *PyTorch*[11]. The computations are conducted on a Linux cluster with 2 Nvidia V100 GPU machines (each with 32 Gb memory) and 32 CPU with a total memory of 100 Gb. We use the *sparse Adam*[12] optimizer to update the hidden factors (user and item embeddings), and the usual Adam optimizer to update the remaining parameters. This is because both the user and item embeddings are relatively sparse in the datasets. Therefore, the Adam algorithm, which leverages the momentum of the gradients from the previous batch, may not be ideal for updating the item and user embedding in the current batch. The sparse Adam optimizer address the above issue.

## C.3 COMPLETE NUMERICAL RESULTS

Table A.1: Additional testing results on MovieLens-1M. The settings of this table follow from Table 1. **DT-(M/M/N)**, for instance, indicates using the proposed method with $f$ and MCF, $w$ as NCF and $g$ as NCF. We highlight the **best** results.

| | **DT**-(M/N/N) | **DT**-(M/M/N) | **DT**-(N/M/M) | DR-MF | CE-MF |
|---|---|---|---|---|---|
| | *MovieLens-1M* | | | | |
| Rel@10 | 14.55 (.11) | **14.63** (.10) | 14.52 (.08) | 14.28 (.12) | 14.17 (.08) |
| Hit@10 | 62.34 (.20) | **63.02** (.21) | 62.18 (.19) | 61.03 (.16) | 61.22 (.18) |
| NDCG@10 | 32.75 (.11) | **33.21** (.23) | 32.90 (.16) | 31.97 (.08) | 32.04 (.07) |
| | *MovieLens Simulation* | | | | |
| Rel@10* | 4.62 (.03) | 4.67 (.02) | **4.84** (.04) | 3.89 (.03) | 3.91 (.02) |
| Hit@10 | **79.89** (.17) | 79.80 (.15) | 79.83 (.20) | 75.99 (.10) | 75.69 (.12) |
| NDCG@10 | 39.98 (.10) | 39.94 (.11) | **40.57** (.11) | 38.38 (.07) | 38.21 (.06) |

We present the complete numerical results in this part of the appendix. We start with the testing performance with respect to the methods for MovieLens-1M that are not shown in the main paper due to the space limitation.

---

[7] https://github.com/hexiangnan/neural_collaborative_filtering

[8] https://github.com/usaito/unbiased-implicit-rec-real for IPW-MF and https://github.com/dawenl/expo-mf for ExpoMF.

[9] https://github.com/StatsDLMathsRecomSys/Adversarial-Counterfactual-Learning-and-Evaluation-

[10] https://github.com/kang205/SASRec

[11] https://pytorch.org/

[12] https://agi.io/2019/02/28/optimization-using-adam-on-sparse-tensors/

Table A.2: Simulation experiment results for the Goodreads simulation. The settings and interpretations are the same as Table 1. We highlight the **best** results.

| | Pop | IPW-MF | ExpoCF | AC-MF | MCF | **DT**-MCF | NCF | **DT**-NCF |
|---|---|---|---|---|---|---|---|---|
| | | | | *GoodReads Synthetic* | | | | |
| Rel@10 | 4.34 (.02) | 4.62 (.09) | 4.57 (.04) | 4.68 (.10) | 4.56 (.06) | **4.78** (.06) | 4.99 (.05) | **5.13** (.07) |
| Hit@10 | 42.75 (.04) | 49.22 (.34) | 48.97 (.26) | 49.30 (.25) | 48.06 (.25) | **49.94** (.21) | 52.47 (.18) | 52.96 (.21) |
| NDCG@10 | 23.20 (.02) | 25.91 (.15) | 25.76 (.17) | 26.04 (.13) | 25.25 (.15) | **26.72** (.12) | 28.17 (.11) | **29.78** (.13) |

Table A.3: Additional numeric results for the variations of the proposed domain transportation methods with different combinations of the $f$, $g$ and $w$ models).

| | **DT**-(M/N/N) | **DT**-(N/M/M) | **DT**-(M/N/N) | **DT**-(N/M/M) | **DT**-(M/N/N) | **DT**-(N/M/M) |
|---|---|---|---|---|---|---|
| | *GoodReads* | | *GoodReads Simulation* | | *LastFM* | |
| Rel@10 | 5.92 (.04) | **6.28** (.05) | 4.65 (.04) | 4.95 (.06) | 6.53 (.06) | 6.38 (.07) |
| Hit@10 | 58.48 (.22) | **61.64** (.30) | 48.97 (.26) | 49.30 (.25) | 81.09 (.33) | 78.96 (.31) |
| NDCG@10 | 30.32 (.17) | 36.32 (.22) | 25.88 (.14) | 28.61 (.13) | 52.05 (.18) | 51.37 (.20) |

**Additional results for MovieLens-1M and its synthetic data**. In Table A.1, we show the additional results for the MovieLens experiments, including using the proposed domain-transportation approach with different configurations of $f$, $w$ and $g$ (denoted by **DT-(f/w/g)**). Note that Attn does not apply to the synthetic data, because the ordering in the synthetic data is randomly generated as we mentioned previously. For notation, we use the short hand **DT-(M/N/N)**, where in the parenthesis are the assignments for $f$, $w$ and $g$, and **M** is short for **MCF**, **N** is short for **NCF**. Also, we experiment with the doubly-robust joint learning approach on MCF (denoted by DR-MF) as proposed by Wang et al. (2019), and the causal embedding approach on MCF (denoted by CE-MF) proposed in Bonner and Vasile (2018). The two methods were proposed for the explicit feedback data, and we observe that while they are still able to improve the performance of MCF in our implict feedback setting, their improvements are much less significant compared with the proposed approach. Also, it appears that the causal embedding approach suffers from high variances, which is due to the workaround when constructing the "uniform exposure" dataset from implicit feedback data.

**Complete results for the LastFM data**. In Table A.3, we show the complete results for the LastFM experiments. We do not conduct the synthetic experiments on the LastFM data, because unlike the other two datasets, the original dataset consists only of implicit feedback. Therefore, neither the relevance nor the exposure can be efficiently estimated from the data. The settings, interpretations and notations for our approach are the same as above.

**Complete results for the Goodreads simulation experiment**. In Table A.2 and Table A.3, we show the complete results for the Goodreads experiments. Similarly, *Attn* does not apply to the synthetic data for the same reason discussed above.

We then present the complete results in correspondence to Figure 2. In Figure A.1, we the analysis of the learnt weights on the **Goodreads** data and its simulation. In Figure A.2, we show the same set of analysis for the **LastFM** dataset. Finally, in Figure A.3, we provide the ablation study and sensitivity analysis on $\lambda$, for both **Goodreads** and **LastFM** dataset.

C.3.1   SENSITIVITY ANALYSIS ON THE OPTIMIZATION SETTING  AND OVERLAPPING

To ensure the convergence of the minimax optimization without being trapped in a cycle, it is a standard practice to use the two-time-scale GDA algorithm. Our situation is even more involved since we not only have a minimax game, but there are two minimization players.

Intuitively, $f$ is moved only when $P_w$ has approximated $P_f$ sufficiently well. It means that $w$ has converged in the sense that $d_W(P_f, P_w)$ is sufficiently small. Since we use the Monge and Kantorovich formulation to represent the 1-Wasserstein distance $d_W$, i.e., $d_W(P, Q) = \sup_{g:\|g\|_L \leq 1} \int_{\mathcal{D}} g dP - \int_{\mathcal{D}} g dQ$, we need to ensure $g$ to converge every time $w$ moves so that the integral difference is approximately equal to $d_W(P_f, P_w)$. In other words, this is a three-layer loops: when $f$ moves one step, we need $w$ to move $\text{step}_w$ steps until convergence; when $w$ moves one step, we need $g$ to move $\text{step}_g$ steps until convergence.

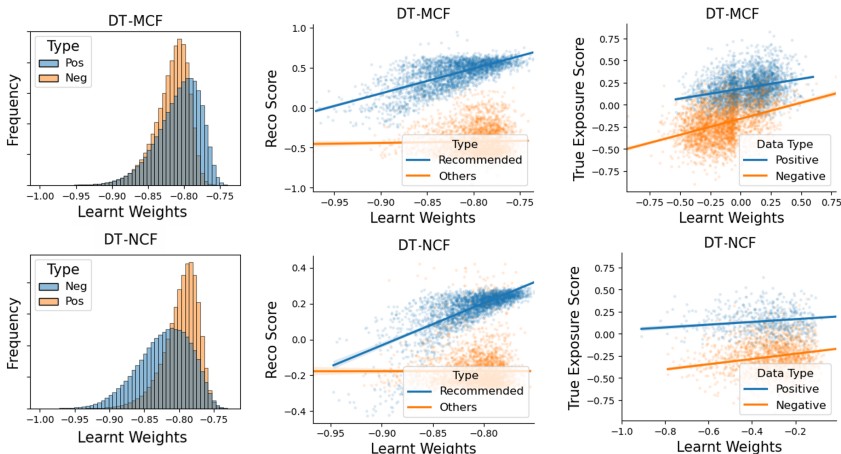

Figure A.1: Analysis on the learnt weights for the **Goodreads** dataset. The setting and interpretation of the figure follows from Fig. 1a and 1b in Section 5.

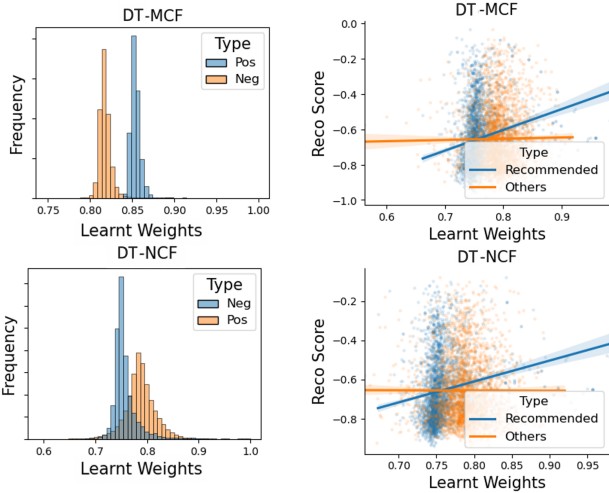

Figure A.2: Analysis on the learnt weights for the **LastFM** dataset. Note that the semi-synthetic experiments are not conducted on the LastFM dataset. The setting and interpretation of the figure follows from Fig. 1a and 1b in Section 5.

Figure A.4a shows that as the $\text{step}_w$ increases when $\text{step}_g$ is fixed, the overall performance of our method is boosted, which aligns with the above claim that $w$ should converge before $f$ moves. We observe similar phenomenon when $\text{step}_g$ increases as $\text{step}_w$ is fixed, which corroborates the claim that $g$ should converge before $w$ moves. These results suggest that we might want to use a large $\text{step}_w$ and a large $\text{step}_g$. However, it can be computationally expensive. From our experience, we can use a relatively large $\text{step}_g$ and a slightly smaller $\text{step}_w$ to achieve a good performance. In addition, appropriate choices of learning rates $lr_f$, $lr_w$ and $lr_g$ can expedite the convergence of $w$ and $g$ so that we can use less steps.

## C.4 ANALYSIS FOR THE COMPLETE RESULTS

We first observe from Table A.1 that the sequential recommendation methods achieve overall better performances than the others. It is expected because the sequential signal in the MovieLens-1M dataset is usually substantial, as point out by Kang and McAuley (2018). Similar to the previous experiments, DT-Attn improves the performance of the regular Attn model. As for the variations of our methods, they all improve the performance of the corresponding $f$ model, and even outperforms

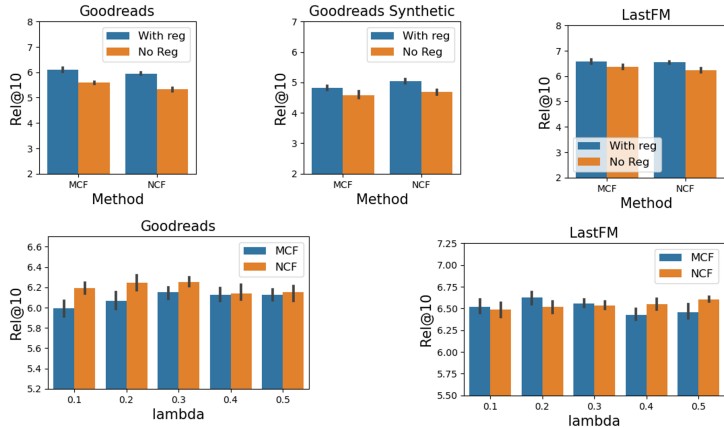

Figure A.3: The ablation study on the regularization component, and the sensitivity analysis on $\lambda$, for both the **Goodreads** and **LastFM** datset. The setting and interpretation of the figure follows from Fig. 1d in Section 5.

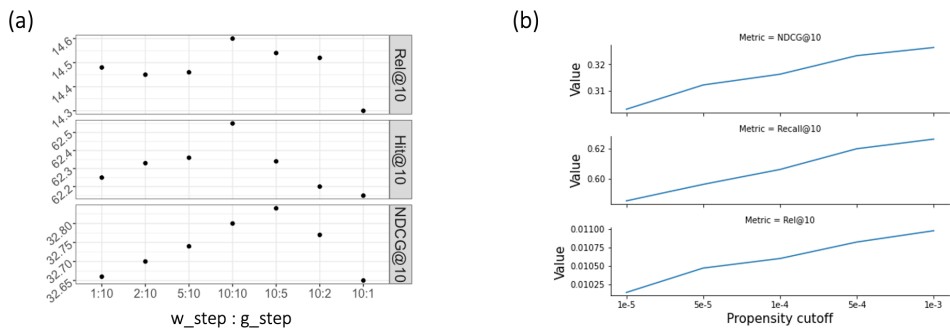

Figure A.4: (a).Performance of minimax optimization with different number of optimization steps for updating $w$ and $g$. The analysis is conducted on the **MovieLens-1M** dataset. (b). Sensitivity analysis on the threshold of propensity score for IPW-MF. In the orignial paper (Saito et al., 2020), the authors apply a threshold on the smallest possible propensity score as if there was sufficient overlap. Here, we vary the threshold to hypothetically generate the different overlapping scenarios.

the DT methods where $f$, $w$ and $g$ use the same "base model". This is a phenomenon that worth future investigation, since the results might indicate the *DT-(f/w/g)* methods combine the strengths of the base models.

The results for the Goodreads and its simulation experiments in Table A.2 and Table A.3 show the similar patterns. Firstly, using the proposed transportation-regularized approach improves the performances of the baseline methods, and the best results are achieved by our **DT-X** methods. Secondly, while the other enhanced MF/CF methods also improve upon the baseline MCF model, but their improvements are much less significant than the DT approach. Thirdly, the *DT-(f/w/g)* methods also achieve more significant improvements compared with the DT-f methods. We point out that the sequential recommendation method (*Attn*) does not particularly suit the dataset, since it is outperformed by the non-sequential methods. Still, *DT-Attn* achieves better performances than *Attn*. Finally, the additional results for the LastFM dataset in Table A.3 have the similar patterns.

As for the analysis on the learnt weights, we observe from Figure A.1 and Figure A.2 that the patterns we discussed in Section 6 for the **MovieLens** dataset and its simulation holds almost the same for the **Goodreads** and **LastFM** datasets. Firstly, the learnt weights for the instances among the feedback data can be distinguished by whether the instance is positive or negative. We observe that the positive instance tends to have larger weights than the negative instances, which means the weighting model is learnt to emphasise the "higher-quality" positive implicit feedback. Secondly,

the learnt weights also get larger for the instances that will be recommended under $f$, and the larger the score of $f$, the higher the weights. It means the weighting function works harder on balancing $P_f$ with $P_w$, which conforms to our design. Lastly, for the simulation, the learnt weights tend to be larger for the negative instances with higher exposure probability, which can be considered the strong negative instances. This is because they are more likely to be exposed but are not clicked.

## C.5 EFFICIENCY ANALYSIS

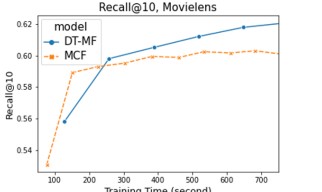 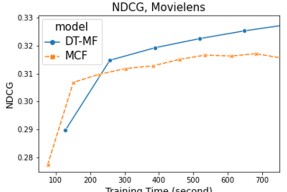 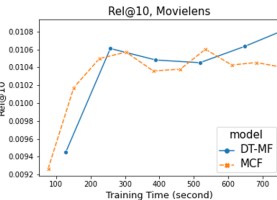

Figure A.5: Analysis on the efficiency of the proposed approach. We report the training time (on the x-axis) and the per-epoch evaluation results (on the y-axis), using the **Movielens-1M** data as examples. We compare between the original MCF and the DT-MF approach, and when training both models, all the model and data configurations are kept exactly the same.

In terms of the space complexity, it is evident that our approach need to store the parameters and gradients of the f, w, and g models (if no parameter sharing is considered). For the time complexity, however, the proposed approach is highly efficient due to the following reasons.

1. As it is shown in Figure A.5, although the DT-MF approach has a larger per-epoch training time, it makes training progress much faster than the original MCF thanks to the rewieghting and DT regularization. For the Movielens dataset, DT-X catches up and outperforms X in a couple of training epochs even before the X model converges.

2. The update of f, w, and g can be efficiently parallelized.

3. Our approach does not change the inference time since only the $f$ component will be used for evaluation and testing, just like when using $f$ alone.

## C.6 DEPLOYMENT TO REAL-WORLD IR SYSTEM AND ONLINE TESTING

An important motivation of our work is to solve the practical challenges of industrial IR systems, and the working mechanism we described in Section 1 applies to most settings. In particular, many real-world recommenders are deterministic, and we can only expose a tiny proportion of the catalog to the customers due to the limited slots and business requirements. For many industrial tasks, the revenue is directly related to the performance of recommender since unqualified recommendations may hurt customer satisfaction severely. Therefore, the feedback data often covers extremely limited support of the product space. It causes a realistic dilemma:

- if the candidate recommender $f$ recommends products very differently from the previous exposure, then the feedback data may not provide enough evidence to assert the performance of $f$;
- if the candidate recommender $f$ recommends products highly similar to the previous exposure, then $f$ will inevitably inherit the various data bias.

From this realistic perspective, our approach aims to find the right balance between the two extremes: the weighting model helps to reshape the feedback data domain (to better fit the objective risk), and the regularizer enforces $f$ to be sufficiently close to the reshaped feedback data domain. To test the competence of our approach for industrial tasks, we apply the proposed domain transportation approach to an industrial recommendation task, and conduct online A/B/C testing to examine the real-world performance.

The context of our task is to recommend items that are similar to the current webpage's product for a major e-commerce platform in the U.S. The platform hosts more than *a hundred million* products

|  | Recall@20 | NDCG@20 | $\hat{\text{CTR}}^*$ |
|---|---|---|---|
| D&W (control) | +0.0 | +0.0 | +0.0 |
| IPW-D&W (variation1) | +0.23% | +0.06% | +0.05% |
| DT-D&W (variation2) | +0.56% | +0.09% | +0.07% |

Table A.4: Offline evaluation of the three models used in the online A/B/C testing. Due to privacy reasons, we are only able to report the relative lift percentage with respect to the baseline. We use D&W to represent the original deep&wide control model, IW-D&W to represent the approximate propensity-weighted version, and DT-D&W to represent the proposed domain transportation approach when coupled with the deep&wide model. *: the $\hat{\text{CTR}}$ is computed using the history exposure log and the past estimated CTR, similar to how we computed the relevance score $Rel$ in the benchmark and simulation experiments.

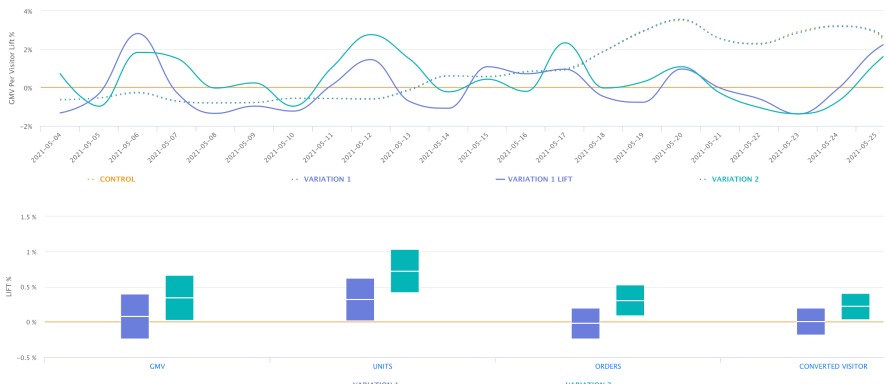

Figure A.6: Online A/B/C testing results for the similar-item model under direct training (**control**), IPW-adjusted training (**variation 1**) and the proposed transportation-constraint risk minimization (**variation2**). The first plot shows the daily gross merchandise value (**GMV**) lift, and the second plot shows the overall lift of *GMV*, number of units per checkout (*UNITS*), number of orders per visit (*ORDERS*), and the total number of converted visitors (*CONVERTED VISITOR*).

ranging from high-end electronics to daily grocery. Each item has its contextual features, including rating, price, taxonomy, etc. The pre-trained embeddings are also available for the items obtained from a modified version of the SGNS algorithm. When applying the item embedding to the model, we use the pre-trained embeddings as initialization and update the parameters with a lower learning rate than the other model parameters. We employ an architecture that is similar to the classical *Deep-and-wide* model, initially proposed by Cheng et al. (2016) for recommending YouTube videos, with the difference that we do not use the real-time features here.

We trained three versions of the deep-and-wide model offline:

- **control**: directly training $f$ using the implicit feedback data;

- **variation1**: first training an exposure model using the past exposure log, apply its normalized version (for controlling the variance) as the weighting function $w$, and then train $f$ in a way similar to the baseline *IPW-MF* mentioned in Section 5;

- **variation2**: train $f$ using the proposed transportation-regularized risk minimization, where $w$ and $f$ are neural CF models that take the item embeddings as input.

All three versions are trained under the same setting. We deploy online A/B/C testing to examine the real-world performances, and the monitored metrics and testing results are provided in Figure A.6. We see that our transportation-regularized risk minimization (variation2) consistently outperforms the original model (control) and the IW-adjusted model (variation1) in term of the gross merchandise value (GMV), which is the most critical metric for industrial recommender system. Also, the proposed approach improves the other monitored metrics (lower panel of Figure A.6) more signif-

icantly than the IW-adjusted model. Our deployment results further suggest the capability of the proposed approach for improving the performance of industrial IR systems.

## D    EXTENDED LITERATURE REVIEW

When viewing our solution from the causal inference perspective, our objective resembles the counterfactual loss proposed by Shalit et al. (2017) focusing on individual treatment effect, which was also discussed in the following work (Johansson et al., 2019; 2020). Further, Johansson et al. (2019) studies from the domain adaptation perspective the issues of insufficient overlapping. Unlike these works where the target domain of interest is fixed in advance, IR systems get to choose the counterfactual world in which they want to act. It leads to the main difference between our work and that line of research: we can control the domain in which the counterfactual risk is computed.

Recently, there has been fruitful research progress in off-policy learning and data missing-not-at-random. Again, most existing methods treat the uniform-exposed domain as the target, while we aim at the deployment domain formed by the IR system. Further, most of them require either a sufficiently randomized logging policy or explicit ratings in the feedback, while our approach is primarily designed for implicit feedback under a deterministic policy. For instance, Sachdeva et al. (2020) studied the overlapping issue for learning from bandit feedback. Similarly, the causal embedding approach (Bonner and Vasile, 2018) requires access to a subset of randomly exposed feedback, and many others rely on the explicit ratings to estimate the historical propensities (Wang et al., 2019; Saito, 2020; Liu et al., 2020). On the other hand, the interventional nature of IR systems is also discussed in Xiao and Wang (2021), but their solution is devised from the reinforcement learning setting.

The Wasserstein distance has also been employed by distribution-robust optimization (DRO) (Kuhn et al., 2019). For instance, Si et al. (2020) uses Wasserstein distance to assert certain robustness to the contextual feature distribution during bandit off-policy learning, and Faury et al. (2020) proposes a robust counterfactual learning framework to handle the various uncertainties in bandit feedback data. Our work differs fundamentally from that research venue as we use Wasserstein distance to regularize learnt policy (via constraining its deployment domain) rather than assert its robustness. Further, DRO methods often have a min-max formulation, while our initial objective in (2) and (3) tackle a min-min problem.

