# OpenReview forum: "From Intervention to Domain Transportation: A Novel Perspective to Optimize Recommendation"
_ICLR.cc/2022/Conference — ICLR 2022 Poster_

### Official Review · Reviewer_n8vS · 2021-11-03

**Correctness:** 3
**Technical Novelty And Significance:** 3
**Empirical Novelty And Significance:** 3
**Recommendation:** 5
**Confidence:** 4

**Main Review:**

The paper is well written and easy to follow. Generally, the overall framework is novel. The authors give some theoretical analyses to the proposed transportation-constraint risk minimization. Extensive experiments are conducted to show the effectiveness of the proposed methods.

My major concerns and questions are as follows:

(1) The motivation is not clear to me. One of the claims of the paper is that the proposed algorithm can address the insufficient overlapping problem. However, the authors do not give a clear motivation of why this problem is important in real-world RL systems and why the proposed methods can address this problem.

(2) The theoretical analyses are not clear and lack interpretations.  Proposition 1 is good to show that IW and DA can not address the insufficient overlapping problem. However, Theorem 1 & 2, Claim 1 and Theorem 3 are not clear.  Specifically,  Theorem 1 and Theorem 2 are only the finite sample bounds under $P$ but not the transfer generalization bound w.r.t the target distribution $Q$ like Proposition 1. It is unclear to me why  Theorem 1 and  Theorem 2 can resolve the insufficient overlapping problem mentioned for using DA in the IR system. For Claim 1, I think the left part in the equation is $\min_{d_{W}(P_{w}, P_{f}) \leq\rho}\mathbb{E}_{P_w} R(f)$

not the risk  $\min_{d_{W}(P_{w}, ~P_{f}) \leq \rho}\mathbb{E}_{Q} R(f)$ under the target $Q$. Thus the Claim 1 is also irrelevant to the target risk.  Theorem 3 indeed gives the generalization bound under the target domain, but it does not clearly show the proposed method can address the insufficient overlapping problem of IW and DA in Proposition 1.  Given above, I think the theoretical part is not well written and it would be great if the authors can provide some interpretations of the theoretical results. The presentation in these parts of the paper could be improved.


(3) The literature review is not very good, and the context is missing about how the research work stands in the agenda of each subfield such as Wasserstein Robust Optimization, Unbiased Recommendation, Off-policy Learning and Causal Inference with Missing data. For instance, the authors miss some important related works [1-7] with deficient support in logging policy and unbiased learning in recommendation. The authors also need to discuss the differences between this paper and other Distributionally robust optimization papers with Wasserstein distance in terms of the generalization bounds. This paper does not make a clear connection with the presented theoretical results and contextualize this work in the existing literature. It is difficult to figure out what's new and what's already a part of the existing literature.

(4) The experimental comparison is also not sufficient. This paper lacks some baselines and results on benchmark datasets.  The comparison should also be done with some DA [8] and advanced IW methods [2]. The authors also need to evaluate their methods on Yahoo and Coat datasets which are the benchmark datasets for counterfactual learning in the recommendation.  A more detailed analysis of the time complexity would be desirable.

[1] Off-policy bandits with deficient support. KDD 2020.

[2] Doubly Robust Joint Learning for Recommendation on Data Missing Not at Random. ICML 2019.

[3] A General Offline Reinforcement Learning Framework for Interactive Recommendation. AAAI 2021.

[4] Distributionally Robust Counterfactual Risk Minimization. AAAI 2020.

[5] Asymmetric Tri-training for Debiasing Missing-Not-At-Random Explicit Feedback. SIGIR 2020.

[6] Distributionally Robust Policy Evaluation and Learning in Offline Contextual Bandits. ICML 2020.

[7] A General Knowledge Distillation Framework for Counterfactual Recommendation via Uniform Data. SIGIR 2020.

[8] Causal Embeddings for Recommendation. RecSys 2018.


**Summary Of The Paper:**

In this paper, the authors study the domain transportation issue in information retrieval (IR) systems. The main challenge is the insufficient overlapping problem. The authors propose a new adversarial learning method based on transportation-constraint risk minimization to address this challenge. The main insight behind this proposed method is that the interventional impact of recommendation is not exclusive to observed data, but also interferes with the target domain of interest. The authors first analyze the generalization bound of IW and DA methods, and then further give the consistency, generalization, and excessive risk bounds of the transportation-constraint risk minimization. The simulation, real-data analysis, and online experimentation show the effectiveness of the proposed method.

**Summary Of The Review:**

 I think this paper is potential. The proposed domain transportation strategy for the recommendation is quite interesting. I suggest the authors focus on this part and give more discussions about the motivation, interpretations, and related works.

---

> ### Author Response · Authors · 2021-11-23
> **We appreciate the comments, and we add experiments, results, extended literature review, and explanations to address the reviewer's concerns**
>
> We thank the reviewer for the careful reading and providing valuable comments on our manuscript. We apologize for the typos and unclear writings. To address the reviewer's concerns, we add several sets of experiments and an extended literature review section as suggested by the reviewer. In the revised manuscript, we mark all the added and modified parts in **RED**). We answer the individual questions as below.
>
>
> ### Further discussion on the motivation
>
> Overlapping has long been recognized as a critical presumption for causal inference and domain adaptation methods. On the theoretical side, we derived in Proposition 1 both the upper and lower generalization bounds associated emphasizing on domain discrepancies, and the bounds reveal how insufficient overlapping can cause arbitrarily poor generalization. We also referred to [1], where the authors use a neural tangent kernel argument to rigorously show that over-parameterized models extrapolate poorly beyond the training domain. To corroborate our theoretical arguments, we add an illustrative experiment in **Appendix C.4** using the Movielens data. We use the fact that IW-MF [2] applies a threshold on the estimated propensity score (e.g. 0.001) to "enforce" a sufficient overlap. We vary the threshold to generate IW-MF's performances under different "degrees of overlapping", and the empirical results are shown in **Figure A.4b** (page 31). It is shown that the performance of IW-MF gets better as we increase the cutoff, which partly suggests insufficient overlapping can cause undesirable performance of recsys.
>
>
> ### Interpretation of the theoretical results
>
> We first apologize for the typo in **Claim 1**: it should be $E_{P_w}$ instead of $E_{P_Q}$ as pointed out by the reviewer.
>
> *Theorem 1* shows that for any policy, the minimal empirical transportation-constraint risk converges to the minimal population transportation-constraint risk at a rate of $1/\sqrt{n}$, so we can safely consider the empirical version.
>
> *Theorem 2* states that the learned policy (the one that minimizes the transportation-constraint risk) is close to the oracle policy in terms of the target risk: the target risk of the learned policy converges to the target risk of the oracle policy at a rate $O(1/\sqrt{n} + R_n(\ell\circ \mathcal{F}))$. Compared with *Proposition 1*, the significance of *Theorem1* and *Theorem 2* is that our learning objective gets rid of the domain discrepancy terms that cannot be controlled.
>
> *Theorem 3* reveals that under the ideal bandit feedback setting with complete overlap, the solution of the proposed transportation-constraint risk minimization is consistent with the solution of the classic counterfactual risk minimization. Therefore, Theorem 3 shows a consistency result that is fundamentally different from the generalization results in *Proposition 1*, so they are not meant to be compared.
>
>
> ### Extending the related-work section
>
> We thank the reviewer for pointing out the additional relevant literature. In the original manuscript, we were unable to include all of them due to the space limitation. Therefore, we decide to add an extended literature review section in **Appendix D** (page 34) to address all the related work in more detail, especially those suggested by the reviewer.
>
>
> ### Additional experiments and results
>
> We manage to add all the experiments suggested by the reviewer:
>
> 1. advanced IPW method -- doubly robust jointly learning [2] -- denoted by **DR-MF**;
>
> 2. domain adaptation method -- causal embedding for recommendation [3] -- denoted by **CE-MF**
>
> 3. the **Yahoo!R3** and **Coat** dataset.
>
> The additional experiment results are shown in **Table A.1, A.2** (page 28), and all the data processing, model configuration, and result interpretation are added to **Appendix C**. We wish to point out that the main reason for not using the Yahoo and Coat data initially is that they do not have timestamps, so the sequential recommendation models cannot be tested.
>
> In general, DT-MF still outperforms all the other (MCF-based) baseline methods, including DR-MF and CE-MF, in all the added experiments including the Yahoo and Coat data.  This is expected because DR-MF, CE-MF, and many other relevant methods are proposed for either the bandit feedback setting or explicit feedback data. The challenge for deterministic and implicit feedback, which is the focus of our work, often lies beyond the comfort zone of those methods, including non-overlapping, inaccurate propensity estimation, and the fact that not all non-clicked data are genuinely negative.
>
> Finally, we would like to thank the reviewer again for the insightful comments, and we are looking forward to future discussions.
>
>
> [1] Xu K, et al, How Neural Networks Extrapolate: From Feedforward to Graph Neural Networks, ICLR'21
>
> [2]. Wang S, et al, Doubly robust joint learning for recommendation on data missing not at random, ICML'19
>
> [3]. Bonner S, et al, Causal embeddings for recommendation, SIGIR'18

---

> > ### Comment · Reviewer_n8vS · 2021-11-23
> > **Interpretation of the theoretical results is not enough.**
> >
> > 1. The motivation is still not very clear. The authors still do not give a clear motivation of why this problem is important in real-world RL systems.
> >
> > 2. Theorem 1 and Theorem 2 are only the finite sample bounds under $P$ but not the transfer generalization bound w.r.t the target distribution like Proposition 1. It is unclear to me why Theorem 1 and Theorem 2 can resolve the insufficient overlapping problem mentioned for using DA in the IR system.
> > Why does Theorem 2 state that the learned policy (the one that minimizes the transportation-constraint risk) is close to the oracle policy in terms of the target risk? Should the target risk be under $Q$?  I can not find any risk on $Q$ in Theorem 1 and 2.
> >
> > 3. Thanks for your additional experiments and results.

---

> > > ### Author Response · Authors · 2021-11-23
> > > **Thanks for clarifying the question, we believe there are two major misconceptions that leads to the confusion**
> > >
> > > We thank the reviewer for the further response that helps to clarify the questions. We believe there are several misconceptions that cause the reviewer's confusion.
> > >
> > > 1. Our work aims only at the IW- and DA-based IR methods where sufficient overlapping is a crucial presumption. We have made our scope very clear in our manuscript that we study the overlapping issues related to those methods,  and we **do not** claim or suggest that all IR methods will suffer from insufficient overlapping. We agree that it would be an interesting topic for follow-up research, but that is certainly beyond the scope of a single paper.
> > >
> > > 2. The fundamental idea of our work is to replace a fixed target domain $Q$ with the controllable deployment domain $P_f$, so clearly, there should **not** be any $Q$ in our main theorems. Also, it is **not true** that the target risk does not appear in our theorems. In the generalization error bound in Theorem 2, the "transported" target deployment risk under $E_{P_{n,f}}$ is obviously on the RHS of the equation. Intuitively, this bound characterizes a tradeoff between interpolation and extrapolation. The "transported" target deployment risk is a consequence of interpolating the observed data and transport (shift) the learnt patterns to $P_f$ for the overlapped part. As for the non-overlapped part, since we make no assumption on the extrapolation performance of the reward model, we pay a price that scales with $\rho$ -- the maximum discrepancy allowed between the observed and deployment domain. Our generalization bound truthfully reflects this tradeoff, and in practice, we treat $\rho$ (reflected by $\lambda$ in our final objective) as the hyperparameter and select the one that gives the best validation performance.
> > >
> > > We hope that we have resolved the reviewer's concerns by clarifying the above misconceptions. Please let us know if there are further questions.

---

### Official Review · Reviewer_j5tz · 2021-11-04

**Correctness:** 3
**Technical Novelty And Significance:** 3
**Empirical Novelty And Significance:** 3
**Recommendation:** 6
**Confidence:** 3

**Main Review:**

Strengths:
* The subject of the paper is quite relevant and the identified issue of non-overlapping support for RW recommendations serious
* The application of existing methods from causal inference to deterministic recommendation policies is valuable and the empirical section is well-developed and the results significant

Weaknesses:
* The motivation for the particular approach is not extremely clear. The authors build the argument around “transportability of patterns”, but I feel the point is never fully explained. If an action is never taken under a certain context in the past, (meaning is in the non-overlapping set), the algorithm will have to extrapolate the outcome. In order for the extrapolation to hold some assumptions need to be made about the reward model, and I find the authors do not really spend time on this.
* The resulting objective is closely resembling the counterfactual loss proposed in [1], eq. 3. I think the authors should expand on this approach in the related work, and highlight that unlike in the cited work, they can choose the counterfactual world in which they want to act. Furthermore, I think borrowing either the counterfactual notation or the policy learning notation and vocabulary would help quite a lot in explaining the approach.





**Summary Of The Paper:**

The authors propose a novel way to learn new recommendation deterministic policies from feedback collected under different recommendation policies.
The authors start by pointing out that the majority of current approaches, either Counterfactual (Inverse Propensity Weighting) either Domain Adaptation-based,
do not work well when the support of the current and future recommendation policies do not fully overlap (one of the main causes being that the current recommendation is deterministic and therefore not exploring all actions).
To this end, the authors propose a novel way of re-weighting and regularizing empirical risk that encourages the search for policies that have better reward (lower risk) and are still overlapping in evidence with the current policy.
To escape the support issue that is present in the existing approaches, the authors switch the regularization parameter to an IPM metric (WASSERSTEIN) that can handle non-overlapping supports.
 The authors provide both theoretical and empirical evidence that the resulting algorithm outperforms existing SOTA methods.

**Summary Of The Review:**

Overall, I liked the paper and I found the approach interesting. I think the authors should link their approach a lot more to the work presented in [1] and also covered in two of their references [2,3].
I think by doing so, it clarifies that the main difference in the problem domain is that we are in control of which of the counterfactual worlds we want to compute the risk in.
Otherwise, all of the vocabulary and the core explanations of the soundness of the approach remain the same.

[1] @inproceedings{shalit2017estimating,
  title={Estimating individual treatment effect: generalization bounds and algorithms},
  author={Shalit, Uri and Johansson, Fredrik D and Sontag, David},
  booktitle={International Conference on Machine Learning},
  pages={3076--3085},
  year={2017},
  organization={PMLR}
}

[2] F. D. Johansson, D. Sontag, and R. Ranganath. Support and invertibility in domain-invariant representations. In The 22nd International Conference on Artificial Intelligence and Statistics, pages 527–536. PMLR, 2019.
[3] F. D. Johansson, U. Shalit, N. Kallus, and D. Sontag. Generalization bounds and representation learning for estimation of potential outcomes and causal effects. arXiv preprint arXiv:2001.07426, 2020.

---

> ### Author Response · Authors · 2021-11-23
> **We appreciate the valuable comments, and we add explanations and an extended related-work section to address the reviewer's concerns**
>
> We thank the reviewer for providing valuable feedback and comments to our manuscript, and we have created an additional literature review section in **Appendix D** (page 34) to include more discussions on the relevant work. In the revised manuscript, we mark all the added and modified parts in **RED**). Here, we address the reviewer's concerns as below.
>
> ### Further discussion on the motivation and the issues with extrapolation
>
> The reviewer made an excellent point that a critical step for learning under non-overlapping is to extrapolate the reward model beyond the observed actions. It is often impossible to guarantee extrapolation performance without making further assumptions, however, our work uses an alternative strategy to resolve this issue as we explain next.
>
> Firstly, our solution makes a major effort to interpolate the observed data via empirical risk minimization. The guarantee for interpolation comes from classical learning theory, with the standard assumption that the actual reward model lies within the $\ell\circ\mathcal{F}$ family we consider.
> Then, the DT constraint critically restricts the future action space where the radius $\rho$ governs the maximum allowance of extrapolating beyond the observed regime. In what follows, we tune the value of $\rho$ (reflected by $\lambda$ in our final objective) to try different extrapolations and select the one that leads to the best empirical validation performance. On the theoretical side, since we make no extra assumption on the extrapolation performance of the reward model, we do pay a price that scales with $\rho$ as we show in the generalization error bound of Theorem 2. Compared with the bounds for IW and DA in Proposition 1, even though the existence of $\rho$ means that "perfect learning" can not be achieved, what matters here is that we can control $\rho$ while the discrepancy terms that trouble IW and DA are essentially out of control.
>
> Therefore, an alternative viewpoint of our motivation is finding a balance between interpolation (what we can guarantee) and extrapolation (what we cannot guarantee), and both our implementation and theoretical justifications cling to this motivation. We apologize for not including the above discussions in the manuscript due to the space limit, and we will add them to the appendix.
>
>
> ### Extending the discussions on relevant literature.
>
> We thank the reviewer for suggesting making connections with the counterfactual learning literature. We add a dedicated section in the appendix to include more discussions on the relevant literature (**Appendix D**, page 34), particularly those listed by the reviewer.
>
> Finally, we would like to thank the reviewer again for the comments, and we are looking forward to future discussions.

---

### Official Review · Reviewer_ozid · 2021-11-09

**Correctness:** 4
**Technical Novelty And Significance:** 3
**Empirical Novelty And Significance:** 3
**Recommendation:** 6
**Confidence:** 3

**Main Review:**

The paper views recommender systems from a new perspective, which is similar but different to existing works in causal collaborative filtering. The theoretical analyses are comprehensive. The experimental studies are conducted in synthetic data, offline data and online data, which can well demonstrate the effectiveness of the proposed method.

I have the following questions/concerns about this paper:
1.	It seems to me that the proposed method can outperform IW methods when the overlap between the source and target domains are insufficient. However, there is no measurement about the insufficiency. It will be better to have some theoretical or empirical analysis about how the overlap between different domains can affect the performance.
2.	It is hard to directly compare the bounds derived by the paper, e.g., between Theorem 3 and Proposition 1. This will be important to intuitively understand the advances of the proposed method.
3.	The online evaluation used the Deep and Wide method. But the same method is not evaluated in the offline setting, which seems unreasonable.
4.	The proposed method tries to optimize recommendation algorithms via a minimax game, which could be much more complicated than other baselines. Thus, it would be necessary to analyze the efficiency of the proposed method in the experiments.


**Summary Of The Paper:**

This paper proposed a new perspective for recommender systems: rephrase optimizing recommendation as finding an intervention that best transports the patterns it learns from the observed domain to its intervention domain. To optimize the recommendation algorithms in this setting, the authors proposed a transportation-constraint risk minimization objective and convert it to a two-player minimax game. Theoretical and empirical studies demonstrated the effectiveness of the proposed method.

**Summary Of The Review:**

In summary, this paper targets at an interesting problem and shows promising theoretical and empirical results, which could be helpful for the community of recommender systems.

---

> ### Author Response · Authors · 2021-11-23
> **Thank you for the feedback, and we add experiments, results, and theoretical explanations to resolve the reviewer's concerns**
>
> We are very thankful for the reviewer's careful reading and providing valuable feedback to our manuscript. We have added experiment results to the revised manuscript as suggested by the reviewer (all the added & modified parts are marked in **RED**), and we address the reviewer's concerns as below.
>
> ### Analysis on how insufficient overlap can affect the performance
>
> Overlapping has long been recognized as a critical presumption for IW methods to work correctly for causal inference and domain adaptation problems. On the theoretical side, we derived in Proposition 1 both the upper and lower generalization bounds associated with IW, and the bounds reveal how the domain discrepancies can cause arbitrarily poor generalization. We also referred to [1], where the authors use a neural tangent kernel argument to rigorously show that over-parameterized models extrapolate poorly beyond the training domain. To corroborate our theoretical arguments, we add an illustrative in Appendix C.4 using the Movielens data. We use the fact that IW-MF [2] applies a threshold on the estimated propensity score (e.g. 0.001) to "enforce" a sufficient overlap. We then vary the threshold to generate IW-MF's performances under different "degrees of overlapping", and the empirical results are shown in **Figure A.4b** (page 31, Appendix). It is shown that the performance of IW-MF gets better as we increase the cutoff, which partly suggests insufficient overlapping can cause undesirable performance.
>
>
> ### Comparing the bounds to understand the advance of the proposed approach
>
> *Theorem 1* shows that for any policy, the minimal empirical transportation-constraint risk converges to the minimal population transportation-constraint risk at a rate of $1/\sqrt{n}$, so we can safely consider the empirical version.
>
> *Theorem 2* states that the learned policy (the one that minimizes the transportation-constraint risk) is close to the oracle policy in terms of the target risk: the target risk of the learned policy converges to the target risk of the oracle policy at a rate $O(1/\sqrt{n} + R_n(\ell\circ \mathcal{F}))$. Compared with *Proposition 1*, the significance of *Theorem1* and *Theorem 2* is that our learning objective gets rid of the domain discrepancy terms that cannot be controlled.
>
> *Theorem 3* reveals that under the ideal bandit feedback setting with complete overlap, the solution of the proposed transportation-constraint risk minimization is consistent with the solution of the classic counterfactual risk minimization. Therefore, Theorem 3 shows a consistency result that is fundamentally different from the generalization results in Proposition 1, so they are not meant to be compared.
>
> ### Offline evaluation of the deep&wide models deployed for online testings
>
> We apologize to the reviewer that due to the company's privacy codes, we can only report the relative percentage lift against the baseline (control model) for our industrial experiments. Therefore, we provided the online testing outcome, which is more straightforward to reveal the genuine performances of the different approaches. In the revised manuscript, we add the offline evaluations in terms of Recall@20, NDCG@20, and estimated CTR in **Table A.5** (page 33, Appendix). In general, we observed that both IW- and DT-enhanced deep&wide model leads to slightly improved offline performances when compared with the baseline deep&wide model, but the improvement from the DT approach is more significant, which agrees with the online testing results.
>
>
> ### Efficiency analysis of the proposed approach
>
> To answer the reviewer's question, we add another section to the revised manuscript (**Appendix C.5** on page 32) to justify the efficiency of our approach. Despite having a minimax training schema, our approach is very efficient for three reasons:
>
> 1. While our DT-X approach has a larger per-epoch training time, the training progress is much more effective than using the original X model thanks to the reweighing and DT regularization. The training time vs. evaluation metric plots in **Figure A.5** (page 33, Appendix) shows that for both the Movielens and Yahoo Music dataset, DT-X catches up and outperforms X in a couple of training epochs even before the X model converges.
>
> 2. The update of f, w, and g can be efficiently parallelized, e.g. $w^{(t)}$ is obtained using $f^{(t-1)}$ and $g^{(t-1)}$. Although we do not directly exploit this acceleration in our implementation, we set it as an option for future experiments.
>
> 3. Our approach does not change the inference time since only the $f$ component will be used for evaluation and testing, just like when using $f$ alone.
>
> Finally, we would like to thank the reviewer again for the comments, and we are looking forward to future discussions.
>
> [1] Xu K, et al, How Neural Networks Extrapolate: From Feedforward to Graph Neural Networks, ICLR'21
>
> [2]. Saito Y, et al, Unbiased recommender learning from missing-not-at-random implicit feedback, WSDM'20

---

### Official Review · Reviewer_Lm9j · 2021-11-11

**Correctness:** 3
**Technical Novelty And Significance:** 3
**Empirical Novelty And Significance:** 3
**Recommendation:** 6
**Confidence:** 4

**Main Review:**

Pros:
	1. This paper identifies and tries to tackle an important problem in recommender systems - insufficient overlapping. The theoretical proof is given to show the uncertainty of the results.
	2. Generally speaking, the paper is well-written and well-presented. Though many mathematical theorems are contained in this paper, it is easy to follow for readers.
	3. The proposed method is novel and can inspire future research in recommender systems.
	4. The experiments are sufficient to validate the claims.

Cons:
	1. There are some small typos that should be fixed before publication.
		1. In claim 1, E_Q R(f) should be E_Pw R(f).
		2. In Table 1, the Hit@10 of DT-MCF on LastFM is 81.81(20). I am confused as 20 is pretty high, and maybe here is (0.20) or (.02).
		3. In Section 5, we randomly sample m < |I| items and see if i' is among the top-K of these m items. items should be unrelated items. Also, the same questions occurred in other parts. I suggest to double-check that.

	2. \mu is also an important hyperparameter, and authors also mentioned that a suitable \mu will give us the opportunity to better explore the whole feedback data. I recommend that the parameter sensitivity of \mu should be conducted.
	3. The main contribution is to address the insufficient overlapping problem that causes the uncertainty of IW and DA-based methods. However, in the experiment part, IW and DA-based methods are not compared with. I recommend that DT-IW and DT-WA should be included.
	4. The motivation is somehow unclear and the presentation of introduction could be further improved. A more detailed explanation about the relationships between intervention and domain adaption, intervention and the proposed method, why the proposed method can address insufficient overlapping problems, etc should be introduced.


**Summary Of The Paper:**

In this paper, the authors first point out that the requirement of efficiency of IW- and DA-based methods is the overlapping between the source domain and target domain. Meanwhile, theoretical analysis also shows that insufficient overlapping problems can cause the hardness or impossibility of IW- and DA-based learning methods. In light of this, this paper aims to propose a novel method to optimize recommendation results by solving such insufficient overlapping problems. Specifically, a principled transportation-constraint risk minimization objective function is devised to optimize the recommendation results, which is able to transport the learned patterns from the source domain to the intervention domain better. Then a two-layer adversarial model (GDA) is deployed to optimize the transportation-constraint risk minimization objective function, and sufficient analysis of the GDA is provided. Extensive experiments were conducted on both real datasets and semi-synthetic datasets to show the superiority of the proposed method.

**Summary Of The Review:**

The novelty of this paper is high.  However, the motivation is not quite clear, and  more IW- and DA-based methods are expected to compare in the experiment.

---

> ### Author Response · Authors · 2021-11-23
> **We appreciate the comments, and we add experiments, empirical results, and explanations to resolve the reviewer's concerns**
>
> We appreciate the reviewer for providing valuable feedback and comments to our manuscript. We have added experiments and arguments to the revised manuscript as suggested by the reviewer (all the added & modified parts are marked in **RED**), and we address the reviewer's concerns as below.
>
> ### Sensitivity Analysis on $\rho$
>
> The sensitivity analysis on $\rho$ (which the reviewer referred to as $\\mu$) is carried out via $\lambda$: according to the duality results in Claim 1 and Eq.(3), the role of $\rho$ (the strength of the constraint) is replaced by $\lambda$ (the strength of the regularization). In **Figure 2(c)** and **Appendix C.3.1**, we conducted extensive sensitivity analysis on $\lambda$ to show that there exists a tradeoff in under- and over-exploring the whole feedback data. Within the range of $\lambda$ we considered, the testing performance experienced moderate variations, and we suggested selecting the hyperparameter via cross-validation.
>
>
> ### Adding IPW and domain adaptation (DA) baselines for comparison
>
> In our original manuscript, we included the propensity-debiased approach [1] as a baseline (denoted by IPW-MF), and in the revised paper, we add two more baselines:
>
> 1. the advanced IPW approach -- doubly robust jointly learning [2] -- denoted by DR-MF;
>
> 2. the DA-based approach -- causal embedding for recommendation [3] -- denoted by CE-MF.
>
> The results for the DR-MF and CE-MF methods are provided in **Table A.1 and A.2** (page 28, Appendix). In general, while the IPW- and DA-based solutions are able to improve the performance of MCF, the proposed domain transportation (DT) approach leads to the best outcomes in all the real-world and simulation data that we experimented with. We point out that the existing IPW- and DA-based recommendations are primarily designed for and tested on the bandit feedback or explicit feedback data where the propensities can be effectively obtained or estimated. We focus on the implicit feedback (which is arguably more widespread and applicable), and the experiments demonstrate that our approach significantly improves upon the existing solutions, including those enhanced by IPW and DA.
>
>
> ### Further Discussion on the Motivation
>
> An alternative viewpoint of our motivation is via Proposition 1 in the Preliminary Section: both the bounds we derived for IW and DA contain terms that explode under a large discrepancy between the source and target domain. They imply that neither the IW-based nor DA-based method may generalize well if there lacks a sufficient overlap. Unfortunately, the non-overlapping issue is intrinsic to many IR and recsys applications, particularly when the historical recommender is deterministic or when the logging policy has limited support.
>
> Our solution seeks to take control of the counterfactual worlds where the future policy is searched. Given the key observation that the intervention from recsys both affects the collected data and forms the target domain after deployment, we attempt to devise a policy that transports the learned patterns on the training domain to its intervention domain. Specifically, such a policy should meet two conditions: (i) its deployment domain has sufficient overlap with the training domain; (ii) it attains the minimal empirical risk when mapped to the training domain (via re-weighting). Our proposed transportation-constraint risk minimization then rigorously formulates the above motivations.
>
>
> ### Clearing the Typos
>
> We apologize for the below typos and unclear writings. We have corrected the typos in the revised manuscript.
>
> 1. in Claim 1, it should be $E_{P_w}$ instead of $E_{P_Q}$ as pointed out by the reviewer;
>
> 2. in experiments, we randomly sample m<|I| **irrelevant** items (so the reviewer's understanding is correct).
>
>
> Finally, we would like to thank the reviewer again for the comments, and we are looking forward to future discussions.
>
> [1]. Saito Y, et al, Unbiased recommender learning from missing-not-at-random implicit feedback, WSDM'20
>
> [2]. Wang S, et al, Doubly robust joint learning for recommendation on data missing not at random, ICML'19
>
> [3]. Bonner S, et al, Causal embeddings for recommendation, SIGIR'18

---

### Decision · Program_Chairs · 2022-01-20

**Decision:**

Accept (Poster)

**Comment:**

This paper presented a domain transportation perspective on optimizing recommender systems. The basic motivation is to view recommendation as applying some form of intervention, implying a distributional shift after the recommendation/intervention. Distribution shift brings tremendous difficulty to traditional causal inference or missing data theory perspective of recommender systems as it violates the distributional overlapping assumption: in simple terms, if the model recommends radically different set of items, there isn't much you can say about its generalization ability; on the other hand, if the model only recommends items that it already observed during training (no distribution shift at all), it would inherent all the biases which already exist in the data. To that end, this paper proposed a domain transportation perspective by introducing a Wasserstein distance constrained risk minimization to find interventions that can best transport the patterns it learns from the observed domain to the post-intervention domain.

The paper received overall borderline scores. All the reviewers acknowledged that the proposed perspective is novel and has the potential to spark a new direction for future work. The reviewers raised concerns, ranging from the bounds in the paper, sensitivity of the optimization w.r.t. the hyperparameter, to some relevant but missing baselines. The authors provided very detailed response and revised the paper quite substantially to address most of the feedback. I also read the paper myself given the borderline scores, and I think the authors did a reasonably good job improving the paper and I agree this paper provides an interesting and novel perspective on viewing recommendation, though I also agree with one reviewer that the idea of "partially extrapolation" can be further explored.

My major complaint is around experimental evaluation. It seems to me that only the semi-synthetic experiment actually makes sense in this context (where the measure is based on the unobserved relevance as opposed to observed click), as the traditional random-split-on-clicks evaluation would inevitably favor models with little distributional shift (the training and test data essentially come from the same distribution, maybe not so with a sequential setting but still close). Furthermore, the inclusion of Yahoo R3 and Coat dataset is even more confusing, as the associated test set implies random exposure which is certainly not what this paper aims to address, unless I am missing something in which case more clarification would be nice.

My overall assessment of the paper is still leaning towards positive but I also wouldn't be too upset if this paper doesn't end up making it. However, if accepted, I do want the authors to carefully revise the presentation of the experimental results for the final version.